# A small, computationally flexible network produces the phenotypic diversity of song recognition in crickets

Jan Clemens[1,2]*, Stefan Schöneich[3,4], Konstantinos Kostarakos[3,5], R Matthias Hennig[6], Berthold Hedwig[3]

[1]European Neuroscience Institute Göttingen – A Joint Initiative of the University Medical Center Göttingen and the Max-Planck Society, Göttingen, Germany; [2]BCCN Göttingen, Göttingen, Germany; [3]University of Cambridge, Department of Zoology, Cambridge, United Kingdom; [4]Friedrich-Schiller-University Jena, Institute for Zoology and Evolutionary Research, Jena, Germany; [5]Institute of Biology, University of Graz, Universitätsplatz, Austria; [6]Humboldt-Universität zu Berlin, Department of Biology, Philippstrasse, Germany

*For correspondence:
clemensjan@gmail.com

**Competing interest:** The authors declare that no competing interests exist.

**Abstract** How neural networks evolved to generate the diversity of species-specific communication signals is unknown. For receivers of the signals, one hypothesis is that novel recognition phenotypes arise from parameter variation in computationally flexible feature detection networks. We test this hypothesis in crickets, where males generate and females recognize the mating songs with a species-specific pulse pattern, by investigating whether the song recognition network in the cricket brain has the computational flexibility to recognize different temporal features. Using electrophysiological recordings from the network that recognizes crucial properties of the pulse pattern on the short timescale in the cricket *Gryllus bimaculatus*, we built a computational model that reproduces the neuronal and behavioral tuning of that species. An analysis of the model's parameter space reveals that the network can provide all recognition phenotypes for pulse duration and pause known in crickets and even other insects. Phenotypic diversity in the model is consistent with known preference types in crickets and other insects, and arises from computations that likely evolved to increase energy efficiency and robustness of pattern recognition. The model's parameter to phenotype mapping is degenerate – different network parameters can create similar changes in the phenotype – which likely supports evolutionary plasticity. Our study suggests that computationally flexible networks underlie the diverse pattern recognition phenotypes, and we reveal network properties that constrain and support behavioral diversity.

## Editor's evaluation

Clemens et al. present a computational model of the cricket song recognition network, which they show is capable of reasonably reproducing neural activity and song selectivity in *G. bimaculatus*. They then explore the parameter space of this network and find that varying parameters of model cells enable it to produce a range of selectivities for the period, pulse duration, duty cycle, or pause duration of input song. They then identify the network parameters that most affect song selectivity and investigate the relationship between several subsets of parameters and song preference. This is a fascinating exploration of the computational flexibility of a small neural circuit; it is well researched and written and was enjoyable to read.

## Introduction

Many behaviors are driven by the recognition and evaluation of sensory stimuli. For instance, hunting requires the detection and tracking of prey; communication requires the recognition of the sounds, pheromones, or visual displays that serve as signals. The diversity of animal behavior across taxa betrays the capacity of sensory systems to evolve and adapt to reliably and specifically recognize a wide variety of signals. Neural evolution is well understood when behaviors are driven by signals recognized through the response specificity of primary afferent neurons, where the change in a single amino acid can change the tuning of a specific behavior (*Prieto-Godino et al., 2017*; *Ramdya and Benton, 2010*). However, many behaviors are driven by complex temporal and spatial signal patterns, whose recognition is based on the processing and comparison of neural activity across time and space, where changes in many parameters define the tuning of the system. For these behaviors, unraveling the underlying neural computation is challenging since it requires a mapping from circuit parameters to recognition phenotype.

One prominent behavior involving the recognition of complex temporal patterns is acoustic communication. Many animals – monkeys, mice, bats, birds, frogs, crickets, grasshoppers, katydids, fruit flies – produce species-specific songs to attract and woo conspecifics of the other sex (*Baker et al., 2019*; *Bradbury and Vehrencamp, 2011*; *Kostarakos and Hedwig, 2014*; *Neunuebel et al., 2015*; *Schöneich et al., 2015*). During the evolution of acoustic communication, the structure of songs as well as behavioral preferences can evolve rapidly during speciation events (*Blankers et al., 2015*; *Mendelson and Shaw, 2005*), giving rise to the large diversity of species-specific songs. Since the evolution of song is mainly driven by the female (*Gray and Cade, 2000*), the females' song recognition must be selective and modifiable in order to drive the evolution of distinct, species-specific song patterns in males (*Wagner, 2007*). But how are these changes implemented at the level of the pattern recognition networks? While electrophysiological experiments can demonstrate the principles of their operations at a given time, they are limited in revealing the functional contribution of cellular and synaptic parameters in a network-wide systematic analysis. To overcome these limitations, the ability of biological networks to generate different recognition phenotypes can be investigated using computational modeling.

Here, we examine the computational capacity of the brain network that recognizes the pulse pattern in the Mediterranean field cricket, *Gryllus bimaculatus*. Cricket song consists of a sinusoidal carrier frequency, modulated in amplitude with temporal structure on short (<100 ms) and long (>100 ms) timescales (*Figure 1A*). On the short timescale, the song consists of trains of sound pulses with a species-specific pulse duration and pulse pause. On the long timescale, the pattern is more variable, and pulse trains are either continuous (trills) or grouped into chirps interrupted by a longer chirp pause. The pulse pattern on the short timescale – and the female's behavioral preference for it – is compactly described in a two-dimensional parameter space spanned by pulse duration and pause (*Figure 1B and C*). The diverse song preferences have been extensively mapped in more than 20 species (e.g., *Bailey et al., 2017*; *Cros and Hedwig, 2014*; *Gray et al., 2016*; *Hennig, 2003*; *Hennig, 2009*; *Hennig et al., 2016*; *Rothbart and Hennig, 2012*). This revealed three principal types of preference, defined by selectivity for specific features of the pulse pattern (*Figure 1B*): pulse duration, pulse period (duration plus pause), and pulse duty cycle (duration divided by period, corresponds to signal energy) (*Hennig et al., 2014*). Intermediates between these types are not known. A fourth type of selectivity – for pulse pause – has not been reported in crickets and is only known from katydids (*Schul, 1998*).

Repetitive patterns of short pulses that are organized in groups on a longer timescale are a common feature of acoustic signaling in insects, fish, and frogs (*Baker et al., 2019*; *Carlson and Gallant, 2013*; *Gerhardt and Huber, 2002*), and the processing and evaluation of these pulse patterns is therefore common to song recognition systems across species. Moreover, circuits analyzing temporal patterns of amplitude modulations are likely building blocks for recognizing the more complex acoustic communication signals found in vertebrates like songbirds or mammals (*Aubie et al., 2012*; *Coffey et al., 2019*; *Comins and Gentner, 2014*; *Gentner, 2008*; *Neunuebel et al., 2015*) including human language (*Oganian and Chang, 2019*; *Neophytou and Oviedo, 2020*). Insights from insects where assumptions on physiologically relevant parameters like synaptic strengths, delays, and membrane properties of individual neurons can be made and systematically tested are therefore relevant for studies of pattern recognition systems and the evolution of acoustic communication systems in general.

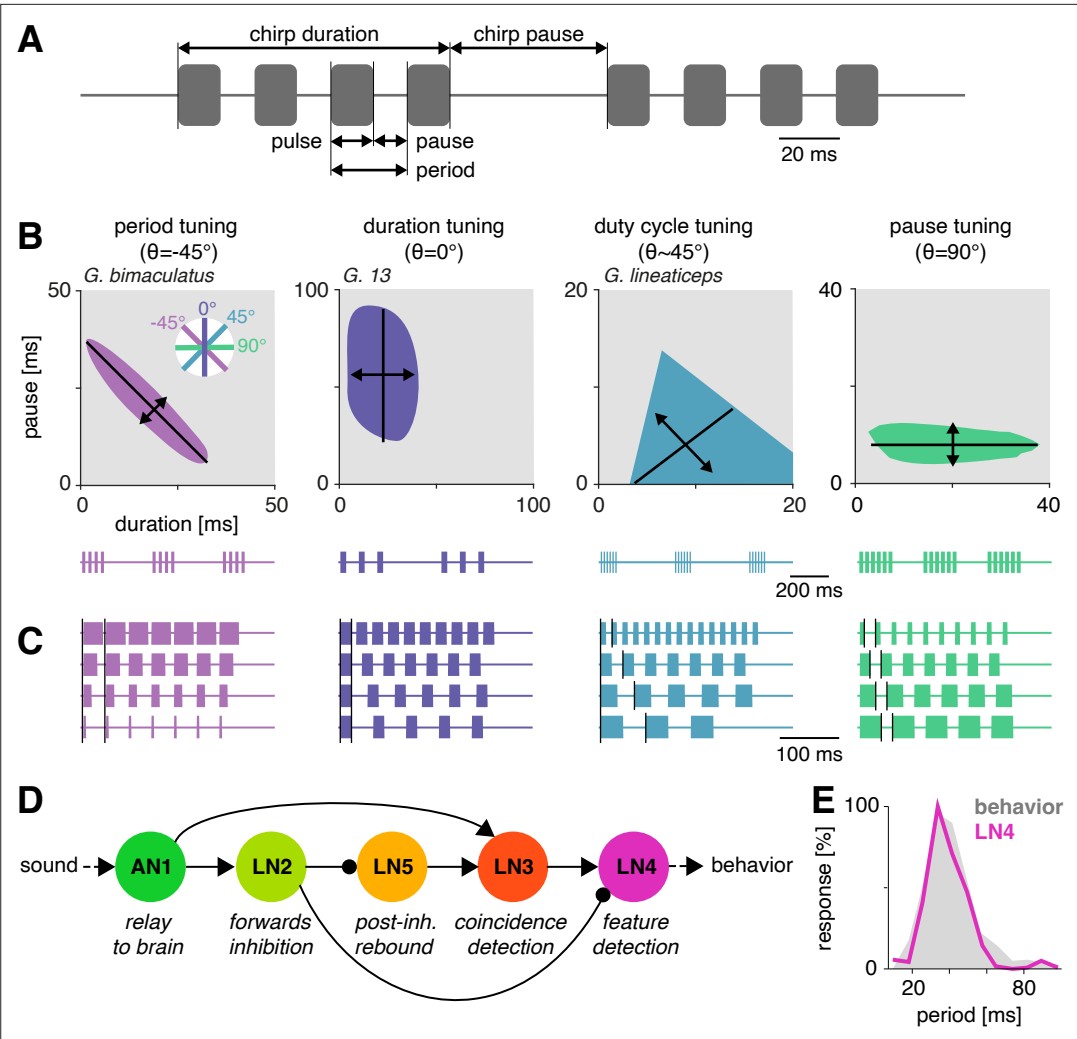

**Figure 1.** Song structure and song preference in crickets, and the song recognition network of *Gryllus bimaculatus*. (**A**) Parameters of the temporal pattern of cricket song. Short sound pulses are interleaved by pulse pauses. The pulse period is given by the sum of pulse duration and pulse pause. The pulse duty cycle corresponds to signal energy and is given by the ratio between pulse duration and period. In many species, pulses are grouped in chirps and interleaved by a chirp pause, while other species produce continuous pulse trains, called trills. (**B**) The behavioral tuning for pulse patterns can be characterized using response fields, which mark the set of behaviorally preferred pulse parameters in a two-dimensional diagram spanned by pulse duration and pause duration. Shown are schematics derived from behavioral data from particular species illustrating the four principal response types known from crickets and other insects. Traces below each response field show typical song patterns for each species. Response types can be defined based on tolerance (black lines) and selectivity (double headed arrows) for particular stimulus parameters, leading to specific orientations of the response field (see left inset): period tuning (purple, *G. bimaculatus*) is defined by selectivity for pulse period and tolerance for pulse duty cycle, giving an orientation of the response field of –45°. Duration tuning (lilac, *G13*, **Gray et al., 2016**) leads to vertically oriented response fields. Duty cycle tuning (cyan, *Gryllus lineaticeps*, **Hennig et al., 2016**) leads to diagonally oriented response fields. Pause tuning (green) with horizontal response fields is not known from crickets but has been reported in the katydid *Tettigonia viridissima* (**Schul, 1998**). The response field given represents a hypothetical cricket species. (**C**) Example stimulus series illustrating the stimulus features each response type in (**B**) is selective for. Vertical black lines mark the feature that is constant for each stimulus series. For duty cycle (cyan), the ratio between pulse duration and period is constant. (**D**) Song recognition network in the brain of *G. bimaculatus*. The network consists of five neurons, each with a specific computational role, which are connected in a feed-forward manner using excitation (pointed arrowheads) and inhibition (round arrowheads). The excitatory ascending neuron 1 (AN1) relays information from auditory receptors in the prothorax to the brain. The inhibitory local neuron 2 (LN2) inverts the sign of forwarded responses. LN2 inhibits the non-spiking LN5 neuron, which produces a post-inhibitory

*Figure 1 continued on next page*

*Figure 1 continued*

rebound. LN3 acts as a coincidence detector for excitatory input from AN1 and LN5. Input delays are tuned such that LN3 is maximally driven by the conspecific pulse train with a pulse period of 30–40 ms. LN4 integrates excitatory input from LN3 and inhibitory input from LN2 and further sharpens the output of the network. (**E**) Tuning for pulse period in LN4 (purple) matches the phonotactic behavior (gray) of *G. bimaculatus* females (**D, E** adapted from Figures 5A and 6A; *Schöneich et al., 2015*).

© 2015, Schöneich et al.  Figure 1D & E are adapted E Tuning for pulse period in LN4 (purple) matches the phonotactic behavior (gray) of *G. bimaculatus* females (D, E adapted from Figures 5A and 6A, *Schöneich et al., 2015*).

While little is known about the neural substrates that recognize song on the long timescale of chirps, the neuronal circuit that computes the behavioral preference for the pulse pattern on the short timescale has been revealed in the cricket *Gryllus bimaculatus* (*Kostarakos and Hedwig, 2012*; *Schöneich et al., 2015*). In this species, the selectivity for a narrow range of pulse periods is created in a network of five neurons and six synaptic connections by combining a delay line with a coincidence detector (*Figure 1D*). The ascending auditory neuron 1 (AN1) is tuned to the carrier frequency of the male calling song and provides the input to a small, four-cell network in the cricket brain. Driven by AN1, the local neuron 2 (LN2) inhibits the non-spiking neuron LN5, which functions as delay line and produces a post-inhibitory rebound depolarization driven by the end of each sound pulse. The coincidence detector neuron LN3 receives direct excitatory input from AN1 and a delayed excitatory input driven by the rebound of LN5; it fires strongly only if the rebound from LN5 coincides with the onset of the AN1 response to the next syllable. Lastly, the feature detector neuron LN4 receives excitatory input from LN3 and inhibitory input from LN2, which sharpens its selectivity by further reducing responses to pulse patterns that do not produce coincident inputs to LN3. LN4's selectivity for pulse patterns closely matches the phonotactic behavior of the females (*Figure 1E*).

We here asked whether the network that recognizes features of the pulse pattern on the short timescale in *G. bimaculatus* (*Figure 1D*) has the capacity to produce the diversity of recognition phenotypes for pulse duration and pause known from crickets and other insects (*Figure 1B*), and what circuit properties support and constrain this capacity. Based on electrophysiological recordings (*Kostarakos and Hedwig, 2012*; *Schöneich et al., 2015*), we fitted a computational model that reproduces the response dynamics and the tuning of the neurons in the network. By exploring the network properties over a wide range of physiological parameters, we show that the network of *G. bimaculatus* can be modified to produce all types of preference functions for pulse duration and pause known from crickets and other insect species. The phenotypic diversity generated by the network is shaped by two computations – adaptation and inhibition – that reduce responses and point to fundamental properties of neuronal networks underlying temporal pattern recognition.

## Results
### A computational model of the song recognition network in *G. bimaculatus*

We tested whether the delay line and coincidence detector network of the cricket *G. bimaculatus* (*Figure 1D*) can be modified to produce the known diversity of preference functions for pulse duration and pause in cricket calling songs (*Figure 1B*). This network was previously inferred from the anatomical overlap together with the dynamics and the timing of responses of individually recorded neurons to a diverse set of pulse patterns (*Kostarakos and Hedwig, 2012*; *Schöneich et al., 2015*). Given that electrophysiology is challenging in this system, dual-electrode recordings to prove the existence of the inferred connections do not exist presently. We consider the neurons in the network cell types that may also comprise multiple cells per hemisphere with highly consistent properties across individuals (*Schöneich, 2020*). We fitted a computational model based on intracellularly recorded responses of the network's neurons to pulse trains. Our goal was to obtain a model that captures the computational capacity of the network without tying it to a specific biophysical implementation, and we reproduced the responses of individual neurons using a phenomenological model based on four elementary computations (*Figure 2A*): (1) filtering, (2) nonlinear transfer functions (nonlinearities), (3) adaptation, and (4) linear transmission with a delay. Nevertheless, all model components have straightforward

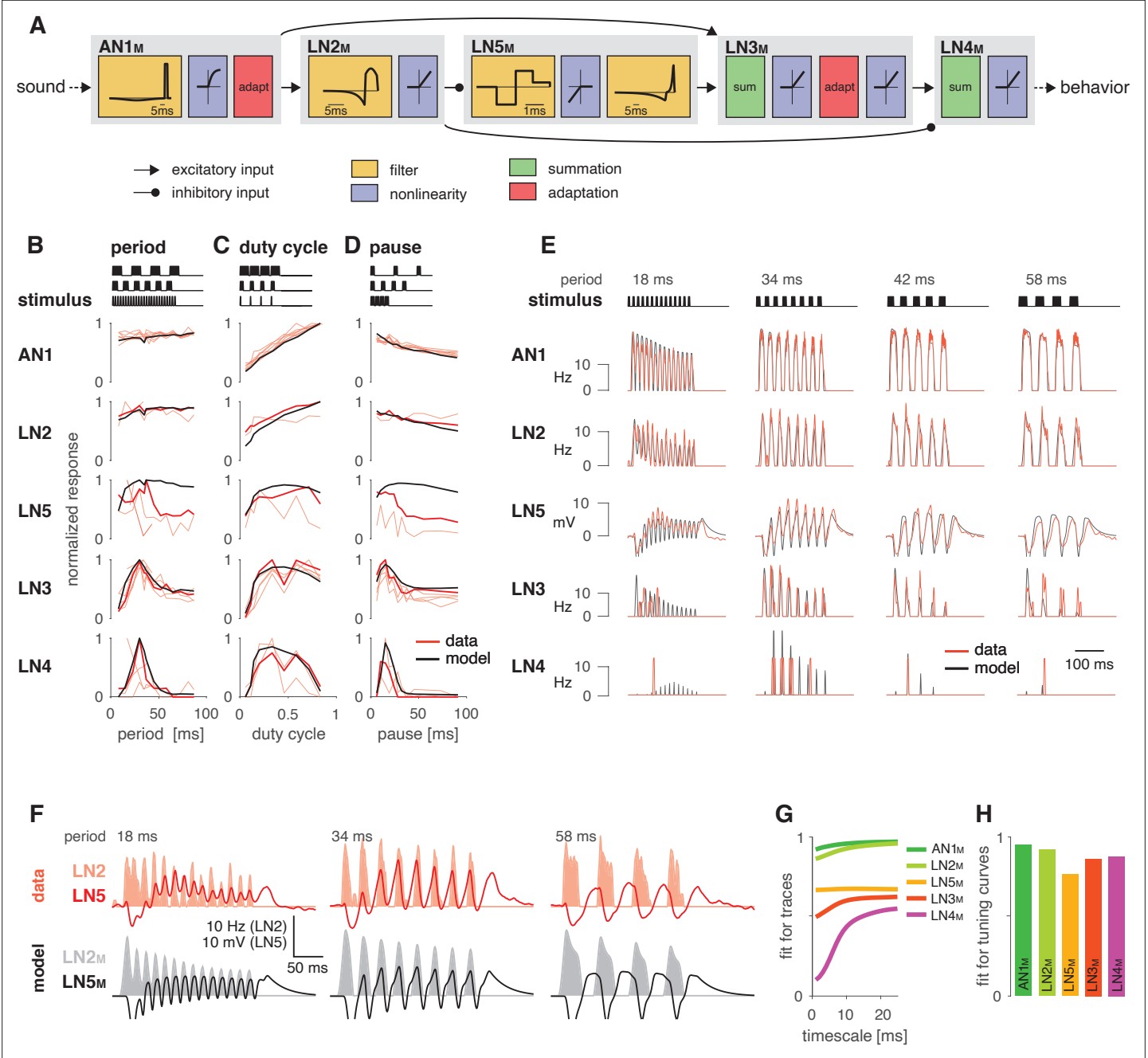

**Figure 2.** A computational model reproduces the responses of the song recognition network. (**A**) The model of the song recognition network (*Figure 1D*) combines four elementary computations: linear filtering (yellow), static nonlinearities (blue), adaptation (red), and synaptic transmission (black lines, pointed arrowheads: excitatory inputs; round arrowheads: inhibitory inputs). Multiple inputs to a cell are summed (green). Pictograms inside of each box depict the shapes of the filters and of the nonlinearities. Y scales are omitted for clarity. See *Figure 2—figure supplement 1*, *Figure 2—figure supplement 2*, and Materials and methods for details. (**B–D**) Tuning for period (**B**), duty cycle (**C**), and pause (**D**) in the data (red, each line is a trial-averaged recording from one individual) and in the model (black) for the five neurons in the network. Thicker red lines depict the recording used for fitting. Stimulus schematics are shown on the top of each plot. Tuning is given by the firing rate (AN1, LN2–4) or integral rebound voltage (LN5) for each chirp and was normalized to peak at 1.0 across the whole set of tuning curves shown in (**B–D**). A duration tuning curve is not shown since it is not contained in the electrophysiological data. See *Figure 3* for duration tuning generated by the model. Number of individual recordings is 8/4/3/6/4 for AN1/LN2/5/3/4. (**E**) Firing rate (Hz) or membrane voltage (mV) traces from the recording used for fitting (red) and from the model (black). Stimuli (top) are pulse trains with different pulse periods. (**F**) Firing rate of LN2 in Hz (shaded area) and membrane voltage traces of LN5 in mV (line) in the recording used for fitting (top, red) and in the model (bottom, black) for short (18 ms), intermediate (34 ms), and long (58 ms) periods. The model reproduces the response timing of LN5 and LN2 responses overlap for short and intermediate but not for long periods. (**G**) Goodness of fit for the response dynamics

*Figure 2 continued on next page*

*Figure 2 continued*

of all neurons at different timescales, quantified as the $r^2$ between the traces in the data and the model. Fits for AN1$_M$, LN2$_M$, and LN5$_M$ are good across all timescales. The fits for LN3$_M$ and in particular for LN4$_M$ increase with timescale (>10 ms) due to the sparse and variable spiking of these neurons (see **E**). (**H**) Goodness of fit for the tuning curves, quantified as 1 minus the root mean-square error between the tuning curves from the data and the model (compare black lines and thick red lines in **B–D**). The curves from the data and the model were normalized by the peak of the curve from the data, to make the measure independent of response scale. Performance is high for all neurons. The weaker match for LN5$_M$ stems from larger model responses for stimuli with long periods or pauses (see **B**, **D**).

The online version of this article includes the following figure supplement(s) for figure 2:

**Figure supplement 1.** Working principle of filters and nonlinearities used as building blocks in the model.

**Figure supplement 2.** Illustration of the model parameters.

**Figure supplement 3.** Goodness of fit for the response dynamics for each neuron in the network model, separated by stimulus.

**Figure supplement 4.** LN4$_M$ response fields for different amplitudes of the LN5$_M$ rebound.

biophysical correlates (see Discussion), which allows us to propose biophysical parameters that tune the network in specific implementations. The computational steps – for instance, whether a neuron had an integrating or a differentiating filter (*Figure 2—figure supplement 2*) – were selected such that each neuron's response dynamics could be reproduced (*Figure 2A*, *Figure 2—figure supplement 2*, *Table 1*). The model parameters were first manually tuned to initial values that reproduced the key properties of each neuron's response and then numerically optimized to more precisely fit each neuron's response dynamics and tuning (*Figure 2B–H*). To simplify fitting, we exploited the feedforward nature of the network: we first optimized the parameters of the input neuron to the network, AN1. Then, we went downstream and fitted the parameters of each neuron's downstream partners while fixing all parameters of its upstream partners for all neurons in the network. Electrophysiological data used for fitting were the time-varying firing rate for the spiking neurons AN1, LN2, LN3, and LN4, and the membrane voltage for the non-spiking LN5 neuron, all in response to periodical pulse trains with different pulse durations and pauses (*Kostarakos and Hedwig, 2012*; *Schöneich et al., 2015*). A detailed description of the model parameters, the data used for fitting, and the fitting procedure are described in Materials and methods, *Figure 2—figure supplement 1*, *Figure 2—figure supplement 2*, and *Table 1*.

## The model faithfully reproduces the neural responses

The fitted model closely reproduced the responses of the network neurons to stimuli from the electrophysiological data set (*Figure 2B–H*). To quantify model performance, we assessed the match in the dynamics and in the tuning between the neuronal and the model responses. First, we computed the squared correlation coefficient ($r^2$) between the recorded and the modeled responses (*Figure 2G*, *Figure 2—figure supplement 3*). We performed this correlation analysis at different timescales of the traces by low-pass filtering responses and predictions with filters of different durations. At shorter timescales, the measure is sensitive to the precise response timing, whereas at longer timescales it reflects the match in the coarse features of the firing rate or voltage dynamics. The $r^2$ value is high across all timescales for the model neurons (indexed with 'M') AN1$_M$, LN2$_M$, LN3$_M$, and LN5$_M$, which respond to a pulse in the biological network with multiple spikes or sustained membrane voltage deflections. By contrast, LN4 produces only a few and irregularly timed spikes during a chirp, and therefore $r^2$ is highest for timescales exceeding the duration of a typical pulse (15 ms) (*Figure 2—figure supplement 3*). Second, we calculated the match between the tuning curves derived from the experimental data and the model (*Figure 2H*). The model excellently reproduced the tuning curves of AN1$_M$, LN2$_M$, LN3$_M$, and LN4$_M$. Performance is lower for LN5$_M$ since the model produced overly strong rebound responses for patterns with long pulse periods and pauses (*Figure 2B and D*). In the electrophysiological data, the rebound amplitude is also variable across individuals. This may reflect interindividual variability, but it could also be an experimental artifact due to the challenges of recordings from the tiny branches of this very small neuron. Despite this variability, the tuning of responses downstream of LN5 – LN3 and LN4 – is not (*Figure 2B–D*). This indicates that the biological network is robust to small changes in rebound amplitude and that it primarily relies on rebound timing. This is well reproduced in our model (*Figure 2E–G*): the response dynamics and tuning for the downstream neurons LN3$_M$ and LN4$_M$ are well reproduced despite the discrepancy in LN5$_M$ rebound amplitude.

**Table 1.** Model parameters.

See Figure 2—figure supplement 1 and Figure 2—figure supplement 2 for an illustration and methods for a definition of all parameters. * marks parameters that were fixed during training (9/55). [†] marks parameters that were fixed during parameter and sensitivity analyses (10/55, Figures 4–6).

| Cell | Component | Parameters |
|---|---|---|
| $AN1_M$ | Filter excitatory lobe | (Gaussian) width $\alpha$=0.0005, duration = 9.88 ms, input delay = 7.41 ms |
| | Filter inhibitory lobe | (Gaussian) width $\alpha$=2.32, gain $\gamma$=0.06, duration N = 184 ms |
| | Nonlinearity | (Sigmoidal) slope = 1.5, shift = 1.5, gain = 5, baseline = −0.5 |
| | Adaptation | (Divisive normalization) timescale $\tau$=3760 ms, strength w = 2.82, offset x0 = 1*[†] |
| | Output gain | Gain=12.8[†] |
| $LN2_M$ | Input from $AN1_M$ | Delay = 0 ms, gain = 0.19 |
| | Filter excitatory lobe | (Gaussian) width $\alpha$=1.07, duration N = 14.2 ms, gain = 0.272 |
| | Filter inhibitory lobe | (Exponential) decay $\gamma$=5.98 ms, duration N = 1000 ms*[†] |
| | Output nonlinearity | (Rectifying) threshold = 0*[†], gain = 1.33 |
| $LN5_M$ | Input from $LN2_M$ | Delay = 8.39 ms, gain = −0.005 |
| | Postsynaptic filter | (Differentiated Gaussian) duration N = 5.0 ms, width $\alpha$=3.5*[†], gain of the excitatory lobe = 1.15 |
| | Postsynaptic nonlinearity | (Rectifying) threshold = 0*[†], gain = 1*[†] |
| | Rebound filter exc. lobe | (Exponential) decay $\gamma$=3.54 ms, duration N = 20.7 ms, gain = 915 |
| | Rebound filter inh. lobe | (Exponential) decay $\gamma$=30.3 ms, duration N = 500 ms*[†], gain = 1718 |
| | Output nonlinearity | (Rectifying) threshold = 0*[†], gain = 3.82 |
| $LN3_M$ | Input from $AN1_M$ | Delay = 7.33 ms, gain = 32.1 |
| | Input from $LN5_M$ | Delay = 3.16 ms, gain = 3.78 |
| | Postsyn. nonlinearity | (Rectifying) threshold = 0.26, gain = 0.014 |
| | Adaptation | (Divisive normalization) timescale $\tau$=39.4 ms, strength w = 0.283, offset x0 = 1*[†] |
| | Output nonlinearity | (Rectifying) threshold = 2.33, gain = 7.68 |
| $LN4_M$ | Input from $LN2_M$ | Delay = 17 ms, gain = −1205 |
| | Input from $LN3_M$ | Delay = 4.87 ms, gain = 401 |
| | Output nonlinearity | (Rectifying) threshold = 738, gain = 0.0052 |

Moreover, we find that altering rebound amplitude within the range of the discrepancy only weakly affects model output (*Figure 2—figure supplement 4*). We can also not exclude that there is a population of multiple LN5-type neurons in each hemisphere and that variability between the individual LN5 neurons reflected in our recordings is averaged in their summed input to LN3. Overall, this shows that despite this small discrepancy between the data and the model, our model well captures the computations of the biological network.

To further assess the model's performance, we examined each model neuron's responses over a wide range of pulse and pause durations that covered the range of song parameters found across cricket species (*Weissman and Gray, 2019*). There exist no electrophysiological data for such a wide range of stimuli, but the behavioral data from *G. bimaculatus* indicate that the neural responses should change smoothly with the song parameters (*Grobe et al., 2012*; *Hennig et al., 2014*; *Kostarakos and Hedwig, 2012*). The responses of all neurons in the model – presented as two-dimensional response fields that depict the response rate for each combination of pulse duration and pause in the set of

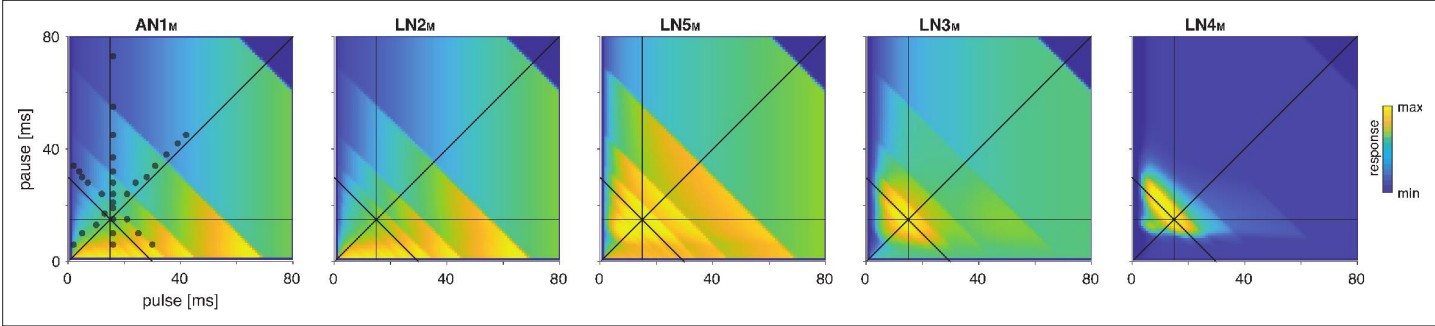

**Figure 3.** Model responses to novel pulse train stimuli. Responses of the model neurons for stimuli with different combinations of pulse and pause durations (1–80 ms, 1600 stimuli per response field, color code corresponds to response magnitude). Each response field depicts the firing rate (for AN1$_M$, LN2$_M$–4$_M$) or the voltage of the rebound (for LN5$_M$) of model neurons. Pulse trains had a fixed duration of 140 ms and were interleaved by a pause of 200 ms, mimicking the chirp structure of *G. bimaculatus* calling song. Anti-diagonal step-like patterns in the response fields arise from changes in the number of pulses per train (*Figure 3—figure supplement 1A and B*). Although the data set used for fitting did not include stimuli with long pulse durations, the model predicts the weak response known from the behavior for these stimuli. Solid black lines indicate stimuli with 15 ms pause duration (horizontal), 15 ms pulse duration (vertical), 30 ms pulse period (anti-diagonal), and 0.5 pulse duty cycle (diagonal). Dots in the leftmost panel mark the stimuli used for fitting.

The online version of this article includes the following figure supplement(s) for figure 3:

**Figure supplement 1.** Stepwise changes in the response fields arise from the stimulus structure.

**Figure supplement 2.** Preference for pulse parameters is robust to small changes in chirp duration.

test stimuli – are consistent with this prediction (*Figure 3A*). Discontinuities in the responses with a stimulus parameter stem from the discrete nature of the stimulus because the number of pulses per train changes with pulse duration and pause (*Figure 3—figure supplement 1A and B*). The response fields illustrate the gradual transformation of tuning in the network: LN2$_M$ at the beginning of the network responds best to stimuli with large duty cycles, that is, stimuli with long pulse durations and short pauses. Following the network from LN5$_M$ over LN3$_M$ to LN4$_M$, the responses to large duty cycle stimuli attenuate and the pulse period tuning becomes more and more prominent, with LN4$_M$ ultimately being selective for a narrow range of pulse periods.

The song of *G. bimaculatus* is produced with a chirp pattern (*Figure 1A*) and female preference for it is broad relative to the tuning for pulse duration and pause (*Grobe et al., 2012*). Consistent with that, the model's response field is robust to the small changes in the chirp duration like adding or removing a single pulse from a chirp typically found in the natural song of this species (*Figure 3—figure supplement 2*). This confirms that our results on pulse duration and pause are robust to the small variations on the longer timescale of chirps observed in natural song. Future studies will examine to what extent the network that recognizes song on the short timescale of pulses also contributes to female preference on the longer timescale of chirps.

In summary, our model reproduces the characteristic response features of each neuron type in the biological network. Using this model of the song recognition mechanism in *G. bimaculatus*, we can now test whether the network has the capacity to produce the behavioral preferences for pulse duration and pause known from other cricket species and identify the parameters that determine the network's preference.

## The network can be tuned to produce all known preferences for pulse duration and pause in crickets

To determine the diversity of behavioral preferences for pulse duration and pause that the network can produce, we created different model variants by altering all model parameters – for instance, the weight and delay of inputs or the amplitude or duration of filters. The model variants were generated by randomly picking values for each of the 45 parameters from an interval around the parameter values of the fit to *G. bimaculatus* (see Materials and methods for details and *Table 1*). Biophysical parameters of a neuron type can vary 10-fold even within a species (*Goaillard et al., 2009*; *Schulz et al., 2007*), and we therefore chose an interval of 1/10 to 10 times the values from the fit to *G. bimaculatus*. Initial experiments with a wider range (1/100 to 100-fold) yielded qualitatively similar results

but with a larger fraction of untuned or unresponsive models. Delay parameters were randomly picked from an interval between 1 and 21 ms. Delay parameters correspond to the delay added to a neuron's inputs and were optimized during fitting to match the timing of the responses of the neuron's outputs. They therefore account not only for axonal transduction and synaptic transmission delays but also for delays arising from low-pass filtering or integration of inputs to the spiking threshold (*Creutzig et al., 2010*; *Zhou et al., 2019*), justifying the extended range of values. To translate responses of the output neuron of the network – $LN4_M$ – into phonotaxis, we used a simple model: the firing rate of LN4 is strongly correlated with the female phonotaxis in *G. bimaculatus* (*Figure 1E*, *Schöneich et al., 2015*), and we therefore took $LN4_M$'s firing rate averaged over a chirp to predict phonotaxis from the model responses. Integrative processes over timescales exceeding the chirp are known to affect behavior in crickets and other insects (*Poulet and Hedwig, 2005*, see also *Meckenhäuser et al., 2014*; *Clemens et al., 2014*; *DasGupta et al., 2014*). We omit them here since they do not crucially affect responses for the simple, repetitive stimuli typical for pulse trains produced by crickets. The preference properties of the network models with randomized parameter sets were characterized for a two-dimensional stimulus space using pulse trains with all combinations of pulse durations and pauses between 1 and 80 ms (*Figure 4A*). We generated 5 million model variants, 9% of these were responsive and selective and used for all further analyses. This low proportion arises because many parameter combinations produce constant output, for instance, if the firing threshold in AN1 is too low or too high.

As a first step towards characterizing the types of tuning the network can produce, we assessed the preferred pulse duration and pause for each of the 450,000 selective model variants (*Figure 4A*). We find that preferences cover the full range of pulse and pause combinations tested (*Figure 4B*). However, the model variants do not cover the preference space uniformly but are biased to prefer patterns with short pulse durations, short periods, and low duty cycles (*Figure 4C*). Peaks at pauses of 0 ms arise from duty cycle-tuned models with a preference for unmodulated stimuli, and peaks at pauses of 80 ms arise from models preferring pauses beyond the range tested here. In conclusion, the network can produce diverse recognition phenotypes, but this diversity is biased towards specific stimulus patterns.

The preferred pulse parameters – duration, pause, and their combinations period and duty cycle – only incompletely describe a network's recognition phenotype. In the next step, we focused a more exhaustive description of the response fields on aspects that have been well described in behavioral analyses. This allowed us to assess the match between the diversity of response fields in the model with the known biological diversity. Behavioral analyses in crickets and other insects (*Deutsch et al., 2019*; *Hennig et al., 2014*; *Schul, 1998*) typically find oriented response fields with a single peak in the two-dimensional parameter space spanned by pulse duration and pause (*Figure 4A*). The vast majority of these fields have an elongated major axis defining stimulus parameters the female is most tolerant for, and a shorter minor axis defining parameters the female is most selective for. Multi-peaked response fields have been associated with a resonant recognition mechanism and have so far only been reported in katydids (*Bush and Schul, 2005*; *Webb et al., 2007*), not in crickets. The orientations of response fields measured in more than 20 cricket species cluster around four angles (*Figure 1B*) forming four principal types of tuning. Intermediate types of phonotactic tuning may exist but have not been described yet. Specifically, duration tuning is defined as selectivity for pulse duration and tolerance for pause (*Figure 1C*, lilac) (*Teleogryllus commodus*, *Gray et al., 2016*; *Hennig, 2003*, see also *Deutsch et al., 2019*). This corresponds to the response field's major axis being parallel to the pause axis (defined as an orientation $\theta$ of 0°, see inset in *Figure 4D*). By contrast, pause tuning (*Figure 1C*, green) corresponds to an orientation $\theta$ of 90°, with the response field's major axis extending parallel to the pulse duration axis. This type of tuning is not known in crickets, only in katydids (*Schul, 1998*). Pulse period and duty cycle tuning correspond to response fields with diagonal and anti-diagonal orientations, respectively. Period tuning (*Figure 1C*, purple) is given by an anti-diagonal orientation ($\theta = -45°$), indicating selectivity for pulse period and tolerance for duty cycle (*G. bimaculatus*, *Hennig, 2003*; *Hennig, 2009*; *Rothbart and Hennig, 2012*). Last, duty cycle tuning (*Figure 1C*, cyan) is given by diagonal alignment ($\theta = 45°$) and selectivity for duty cycle but tolerance for period (*Gryllus lineaticeps*, *Hennig et al., 2016*).

We first examined to what extent the model produced the single-peaked, asymmetrical response fields typical for crickets. We find that most response fields (80%) produced by the selective model variants were well described by a single ellipse (*Figure 4—figure supplement 2A and B*, see Materials

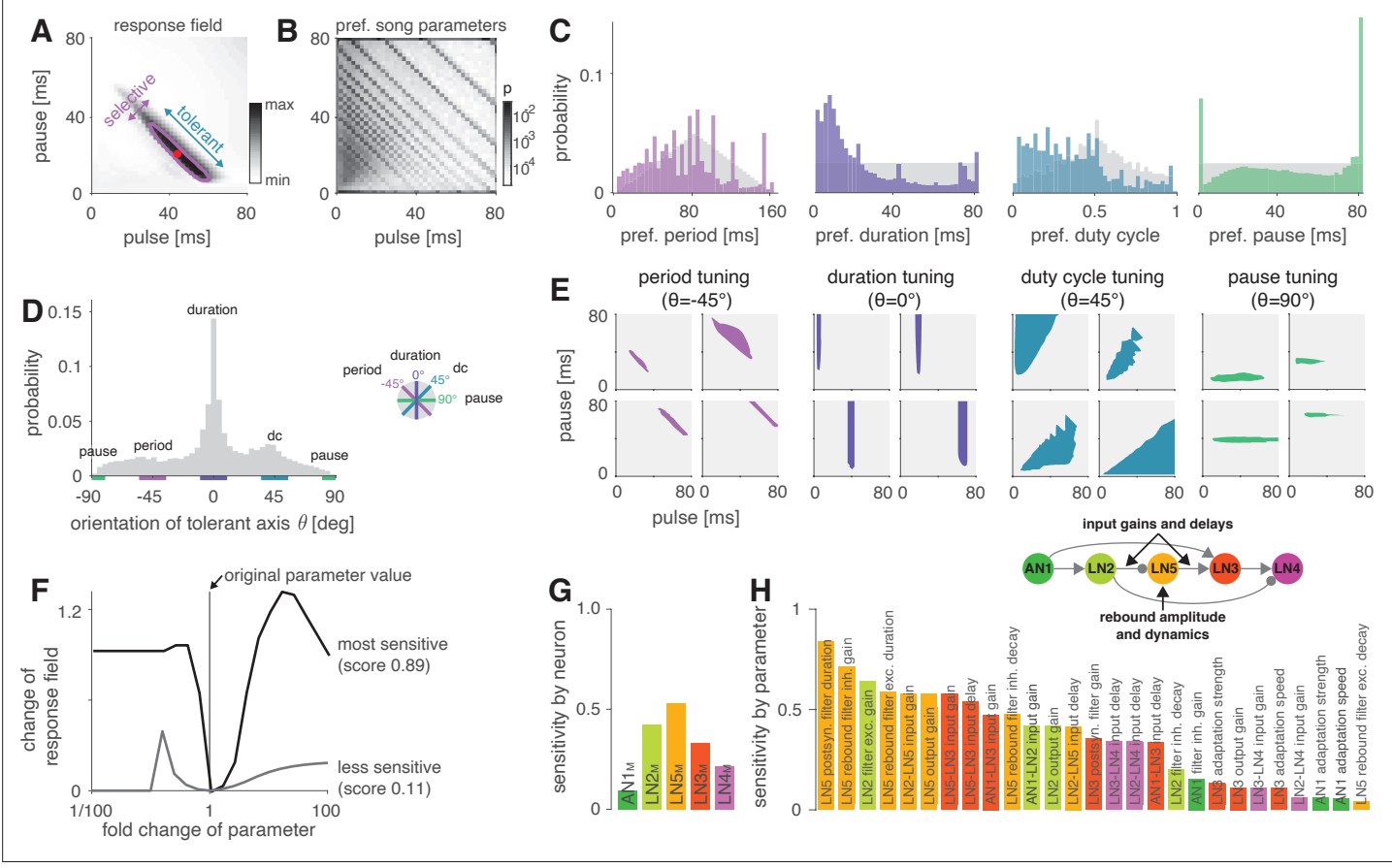

**Figure 4.** The network generates the diversity of response profiles known from crickets and other insects. (**A**) Response field generated from a model variant with randomized parameters. Response magnitude coded in gray colors (see color bar). Tuning was characterized in terms of the preferred pulse and pause durations (red dot) and as tolerant (blue) or selective (purple) directions in the stimulus field. This example is period tuned (purple contour marks the 75% response level) and the set of preferred stimuli is oriented at –45° (see inset in **D** for a definition of the angles), corresponding to selectivity for period (purple) and tolerance for duty cycle (cyan). (**B**) Distribution of preferred pulse and pause parameters for all model variants generated from randomized parameter combinations (coded in gray colors, see color bar). Anti-diagonal patterns arise from the discrete nature of pulse trains (*Figure 3—figure supplement 1A and B*). Models that prefer pauses of 0 ms correspond to models that prefer constant tones. Enrichment of models that prefer the maximally tested pause of 80 ms indicates that the network can generate preference for longer pauses than tested. Preferences cover the stimulus field. (**C**) Distribution of preferred pulse parameters (left to right: pulse period, pulse duration, duty cycle, and pause). Gray histograms correspond to the distributions expected from uniform sampling of stimulus space – deviations from this distribution indicate response biases. The network is biased to produce preferences for short pulse periods, short pulse durations, and low duty cycles. Peaks in the histograms arise from the discrete nature of pulse trains (*Figure 3—figure supplement 1A and B*) or from boundary effects (see **B**). (**D**) Distribution of the orientation of the response fields (see **A**) for model variants that are well fitted by an ellipsoid, have a single peak, and are asymmetrical. Colored lines indicate the range of angles (±10°) that correspond to the four principal response types (see inset and *Figure 1B*). The network can produce response fields at all angles, including the four principal types of tuning for period, duration, duty cycle, and pause. Response fields with small angles around 0°, corresponding duration tuning, occur most often. (**E**) Examples of tuning profiles for pulse period, duration, duty cycle, and pause. Profiles for all tuning types cover the examined stimulus space. (**F**) To identify model parameters useful for controlling network tuning, we modified each model parameter between 1/100 and 100-fold and calculated the change in the response field. The sensitivity score quantifies how much changing a parameter's value changes the response field. Examples shown are the parameters with the highest (LN5ₘ postsynaptic filter duration, black) and lowest non-zero sensitivity (LN5ₘ rebound filter excitatory decay, gray) (see **H**). (**G**) Average sensitivity scores by neuron. LN5ₘ has the highest score, it most strongly shapes network tuning, consistent with the rebound and coincidence detection being the core computational motif of the network. (**H**) Model parameters ranked by sensitivity score. Parameters that induce no or only a single step-like change in the response field were excluded. Color indicates cell type (same as in **G**). Parameters of LN5ₘ (bright orange) and LN3ₘ (dark orange) rank high, demonstrating the importance of the rebound and coincidence detection for shaping model tuning. The model schematic (inset) highlights the most important types of parameters.

The online version of this article includes the following figure supplement(s) for figure 4:

**Figure supplement 1.** Comparison of sensitivity scores for changes of single and pairs of parameters.

**Figure supplement 2.** Most responses fields are well fitted by ellipsoids, some are multi-peaked.

**Figure supplement 3.** Response fields of all neurons in the network for examples from *Figure 4E*.

and methods for details). Of these, 83% were asymmetrical (major axis >1.25× longer than minor axis), 17% were symmetrical (*Figure 4—figure supplement 2C*). 12% of all models produce multi-peaked response fields (*Figure 4—figure supplement 2D*), which are only known from katydids (*Webb et al., 2007*). The remaining 8% of the response fields were not well described by ellipses and/or did not have multiple distinct peaks. Thus, while the model produces more diverse responses – including complex, multi-peaked ones – most responses do match those typical for crickets.

We next assessed the orientation of the single-peaked, asymmetrical response fields to test to what extent they fall into four principal types (*Figure 4D*). We find response fields with any orientation, again demonstrating that the network can produce more diverse response fields than has been reported in crickets. However, the orientations are unevenly distributed and are enriched for the principal types known from crickets: 36% of the response fields have an orientation of 0 ± 10°, which corresponds to duration tuning (expectation from uniform distribution: 20°/360° = 5.6%). Duty cycle tuning (45 ± 10°) and period tuning (–45 ± 10°) are also enriched, with 17% and 12%, respectively. Notably, pause tuning (90 ± 10°) is not known in crickets and is the only principal type that is rarer than expected from a uniform distribution of orientations (2.0% vs. 5.6% expected). The rarity of pause tuning is consistent with the bias to prefer short pulse durations observed above (*Figure 4C*) since orientations around 90° require response fields that extend parallel to the duration axis. Note that these trends do not depend critically on the ranges of angles chosen for specifying the different response types.

Overall, the response fields generated by the model are roughly consistent with the behavioral diversity in crickets: most response fields form a single, elongated ellipse, similar to the behaviorally measured response fields. Duration, duty cycle, and period tuning are frequent in the models and in crickets, pause tuning is rare in the models and absent in crickets. Interestingly, the network tends to create a larger diversity of response fields than is known from crickets, for instance, fields that are symmetrical, multi-peaked, or have intermediate orientations. This suggests that biases in the network – like the rarity of pause tuning – constrain the distribution of preferences that evolution can select from, and that additional factors – like robustness to noise or temperature – then determine the ultimate distribution of phenotypes. Our analysis of different model variants suggests that this song recognition network can produce all known preference types for pulse duration and pause over the range of stimulus parameters relevant for crickets. This phenotypic flexibility implies that the network may form the basis for the diversity of song recognition phenotypes. We therefore sought to identify model parameters that support that diversity, that is, parameters that change the preference for pulse period or that switch the preference from one type to another. We also looked for parameters that constrain the diversity, for instance, parameters that induce a bias towards low duty cycles (*Figure 4B–D*).

## Post-inhibitory rebound properties and coincidence timing are key parameters that shape preferences

To determine key parameters that control the network tuning and to identify the computational steps that induce the preference bias, we systematically examined the effect of changing individual model parameters on the response fields. We swept each parameter individually in 21 log-spaced steps over an interval of 1/100 to 100-fold around the value from the original fit (*Figure 4F*). We then calculated a sensitivity score for each parameter as the average change in the response field of the network's output neuron, $LN4_M$, over the parameter sweep (see Materials and methods). Parameters that when changed produced mostly unselective or unresponsive models were excluded from subsequent analyses, as were parameters that only induced one or two sudden changes in the response fields. For instance, parameters that control the firing threshold of AN1 were excluded because they turn the input to the network on or off – this produces a large, step-like change in the response field and many unresponsive models. Our sensitivity analysis thereby focuses on parameters suitable for controlling the network's tuning, that is, whose change induces smooth shifts in the model responses while retaining responsiveness and selectivity.

The topology of the pattern recognition network is defined by five neurons (*Figure 1D*). As a first step, we sought to evaluate the importance of each neuron for controlling the network tuning by averaging the sensitivity scores for the parameters of each neuron (*Figure 4G*). This revealed that the network tuning can be least controlled through the parameters of $AN1_M$, the input neuron of the

network, and best controlled through the parameters of the non-spiking neuron LN5$_M$, which generates the delayed post-inhibitory rebound. AN1$_M$ is unsuitable for control because changes in most parameters of AN1$_M$ will not induce a gradual change in tuning but will quickly produce no or unselective responses in the network. The importance of LN5$_M$ is consistent with the idea that the rebound and coincidence detection form the core computations of the network (*Schöneich et al., 2015*) since the dynamics of the rebound response directly influence what stimulus features result in simultaneous inputs to the coincidence-detector neuron LN3$_M$.

To identify important parameters for tuning the network, we ranked them by their sensitivity score (*Figure 4H*). In line with the above analysis for each neuron, the top-ranked parameters directly affect the timing and the amplitude of inputs to LN3$_M$. Among these are the delays and gains for the connections upstream of LN3$_M$ (AN1$_M$→LN3$_M$, LN2$_M$→LN5$_M$, LN5$_M$→LN3$_M$) but also the gain of the excitatory lobe of the LN2 filter (see *Figure 2—figure supplement 1*, *Figure 2—figure supplement 2*). Another group of important parameters affects the dynamics and amplitude of the rebound response in LN5$_M$: first, the duration of the 'postsynaptic' filter of LN5$_M$ (*Figure 2—figure supplement 2*), which is required to reproduce the adapting and saturating dynamics of the inputs to LN5$_M$, visible as negative voltage components in recordings of LN5 (*Schöneich et al., 2015*). Second, the gain of the inhibitory and the duration of the excitatory lobe of the rebound filter that produces the post-inhibitory rebound (*Figure 2—figure supplement 1C*, *Figure 2—figure supplement 2*). A modified sensitivity analysis, in which we changed combinations of two parameters at a time, produced a similar parameter ranking, confirming the robustness of these results (*Figure 4—figure supplement 1*).

Our sensitivity analysis revealed key parameters that change the tuning of the network, but did not address their specific effect, for instance, on the preference for specific pulse durations or periods (*Figure 4E*). This is however crucial for understanding which network parameters need to be modified to produce a specific phenotype and where the bias for stimuli with low duty cycles (short pulses and long pauses) arises (*Figure 4B–D*). We therefore examined the specific effects of some of the top-ranked parameters (*Figure 4H*) on the tuning of the network.

## Relative timing of inputs to the coincidence detector controls pulse period preference

Four of the important parameters identified in our sensitivity analysis affect the timing of inputs to the coincidence-detector neuron LN3. Three of these parameters are the delays of the AN1$_M$→LN3$_M$, the LN2$_M$→LN5$_M$, and the LN5$_M$→LN3$_M$ connections. The fourth parameter – the duration of the filter that shapes input adaptation in LN5$_M$ – also affects the input delays to LN3$_M$ (*Figure 5—figure supplement 1*). The delay between the spikes from AN1$_M$ and the rebound from LN5$_M$ in LN3$_M$ strongly affects network tuning since it determines which pulse train parameters produce coincident inputs required for driving spikes in LN3$_M$ (*Figure 5A*). Increasing this delay – for instance by delaying the rebound from LN5$_M$ – increases the preferred pulse period in LN3$_M$ (*Figure 5B*). This delay was hypothesized to be the core parameter that tunes *G. bimaculatus* to a pulse period of 30–40 ms (*Schöneich et al., 2015*), and our sensitivity analysis identifies this parameter as crucial for shaping the network's tuning.

Interestingly, changing the rebound delay has differential effects in LN3$_M$ and in the output neuron of the network, LN4$_M$. In LN3$_M$, increasing the rebound delay changes both duration and pause preference and increases the preferred pulse period without changing the duty cycle preference (*Figure 5C and D*). However, in LN4$_M$, a longer rebound delay only affects pause preference, but not duration preference, and thereby reduces the preferred duty cycle from 0.50 to 0.25 (*Figure 5D*, *Figure 5—figure supplement 2*). This reduction of the preferred duty cycle is a correlate of the low duty cycle bias observed in the network (*Figure 4C*). We therefore investigated the origin of this effect more closely.

LN4$_M$ receives excitatory input from LN3$_M$ and inhibitory input from LN2$_M$ and applies a threshold to the summed inputs (see *Figure 2A*, *Figure 2—figure supplement 2*). To determine which computation in LN4$_M$ reduces the preferred duty cycle, we removed the inhibition from LN2$_M$ and the threshold in LN4$_M$'s output nonlinearity. While the threshold has only minor effects on tuning, removing the inhibition is sufficient to restore the preference for intermediate duty cycles in LN4$_M$ (*Figure 5B–D*, blue). This implies that the inhibition from LN2$_M$ suppresses the responses to high duty cycles in LN4$_M$. We find that changes in the strength and in the timing of excitatory inputs from LN3$_M$ to LN4$_M$ contribute to this suppression (*Figure 5E and F*). First, the excitatory inputs from LN3$_M$ weaken with increasing

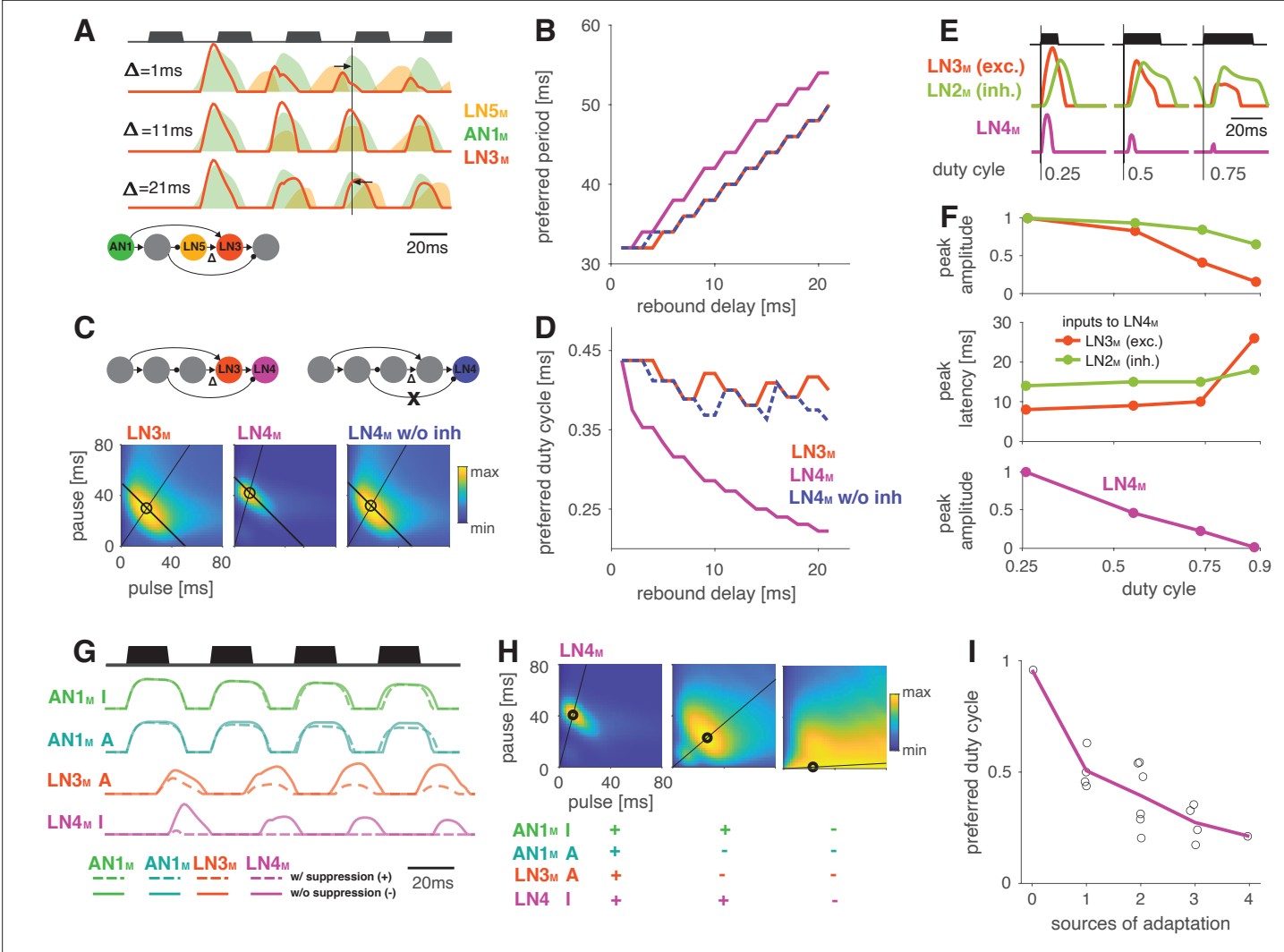

**Figure 5.** Input delays and response suppression control period and duty cycle preference. (**A**) Inputs from LN5$_M$ and AN1$_M$ (orange and green shaded areas) to LN3$_M$ and output of LN3$_M$ (red line) for three input delays from LN5$_M$ to LN3$_M$ ('rebound delay' Δ). The rebound delay is defined as the delay added to the output of LN5$_M$ in the model. The effective delay between the AN1$_M$ and LN5$_M$ inputs to LN3$_M$ depends on the pulse pattern (black, top, pulse duration 20 ms, pause 18 ms). An intermediate delay of 11 ms produces the most overlap between the AN1$_M$ and LN5$_M$ inputs for that stimulus and hence the strongest responses in LN3$_M$. Vertical black line marks an AN1$_M$ response peak, arrows point from the nearest LN5$_M$ response peak. (**B**) Preferred periods for LN3$_M$ (red), LN4$_M$ in an intact model (purple), and LN4$_M$ in a model without inhibition from LN2$_M$ to LN4$_M$ (blue) as a function of the rebound delay. The preferred period increases with rebound delay for all three cases. (**C**) Response fields for LN3$_M$ (left), LN4$_M$ in an intact network (middle), and for LN4$_M$ in a model without inhibition in LN4$_M$ from LN2$_M$ (right) (color coded, see color bar). The rebound delay was set to 21 ms, which increases the preferred period in both LN3$_M$ and LN4$_M$ to 50 ms (left, compare **B**). However, increasing the delay also decreases the preferred duty cycle in LN4$_M$ (middle). Removing the inhibition from LN2$_M$ in LN4$_M$ abolishes the change in duty cycle preference (right). Anti-diagonal lines mark the preferred period of 50 ms for each response field, and lines starting at the origin mark the preferred duty cycle. (**D**) Same as (**B**) but for the preferred duty cycle. With increasing delay, the preferred duty cycle for LN4$_M$ approaches 0.25 but is stable for LN3$_M$ and for LN4$_M$ without inhibition (***Figure 5—figure supplement 2***). (**E**) Inputs to LN4$_M$ (middle, green: inhibition from LN2$_M$; red: excitation from LN3$_M$) and output of LN4$_M$ (bottom, purple) for the intact network in (**C**) and for three different stimulus sequences with a pulse period of 54 ms and increasing duty cycles (top, black). Responses are shown for the second pulse in a train. Excitatory input from LN3$_M$ is weaker and overlaps more with the inhibition for high duty cycles (compare amplitude and latency of response peaks in LN3$_M$), leading to a reduction in LN4$_M$ responses with increasing duty cycle. Y-scales are identical for all three panels and were omitted for clarity. (**F**) Dependence of peak amplitude (top) and peak latency (time from pulse onset to response peak, middle) of inputs to LN4$_M$ (red: excitation from LN3$_M$; green: inhibition from LN2$_M$) on pulse duty cycle for the intact network in (**C**). Weaker and later excitation suppresses LN4$_M$ responses for pulse trains with high duty cycles (bottom, purple). (**G**) Four sources of suppression in the network: the inhibitory lobe in the filter of AN1$_M$ (green), adaptation in AN1$_M$ (cyan) and LN3$_M$ (red), and inhibition in LN4$_M$ from LN2$_M$ (purple). Shown are responses to a pulse pattern (top black, 20 ms pulse duration and 20 ms pause) when the source of suppression is present (dashed lines) or absent (solid lines). Removing suppression produces stronger or more sustained responses. 'A' and 'I' refer to adaptation and inhibition, respectively. (**H**) Response fields (color

*Figure 5 continued on next page*

*Figure 5 continued*

coded, see color bar) for the network output (LN4$_M$) after removing different sources of suppression. The presence or absence of different sources of suppression is marked with a '+' and a '−', respectively. Removing suppression in the network increases the preferred duty cycle. Lines mark the preferred pulse duty cycle, and black dots indicate the preferred pulse duration and pause. (**I**) Preferred duty cycle in LN4$_M$ as a function of the number of sources of adaption present in the model. Black dots show the preferred duty cycle of individual model variants, the purple line shows the average over models for a given number of adaptation sources. Adaptation decreases the preferred duty cycle (Pearson's $r = 0.78$, $p=3 \times 10^{-4}$). See *Figure 2— figure supplement 1* for details. The pulse trains for all simulations in this figure had a duration of 600 ms and were interleaved by chirp pauses of 200 ms to ensure that trains contained enough pulses even for long pulse durations and pauses. Rebound delay set to 21 ms in (**C**) and (**E–I**) to make changes in the duty cycle preference more apparent.

The online version of this article includes the following figure supplement(s) for figure 5:

**Figure supplement 1.** Increasing the duration of the postsynaptic filter in LN5$_M$ delays rebound responses and increases the preferred pulse period in LN3$_M$.

**Figure supplement 2.** Preferred pulse and pause parameters for LN3$_M$ (red), LN4$_M$ (purple), and LN4$_M$ without inhibition from LN2$_M$ over the range of rebound delays tested in *Figure 5B and D* (1–21 ms).

**Figure supplement 3.** Adaptation decreases the preferred duty cycle and has weak effects on the preferred period.

duty cycle, leading to a relatively stronger impact of the inhibition from LN2$_M$ on the responses of LN4$_M$ (*Figure 5F*). Second, the excitatory inputs from LN3$_M$ arrive later with increasing duty cycle, resulting in a more complete overlap with the inhibition from LN2$_M$ and therefore to a more effective suppression of LN4$_M$ spiking responses (*Figure 5E and F*).

These results demonstrate that response suppression by inhibition in LN4$_M$ becomes more effective at high duty cycles and is one source of the network's bias towards low duty cycle preferences (*Figure 4C*). We reasoned that other sources of response suppression, like inhibition or response adaptation elsewhere in the network, could further contribute to this bias.

## Mechanisms of response suppression control duty cycle preference

Four additional computational steps in the network could contribute to the bias against high duty cycles (*Figure 5G*): first, the broad inhibitory lobe of the filter in AN1$_M$ (*Figure 2A*, *Figure 2—figure supplement 1*, *Figure 2—figure supplement 2*) reduces responses to subsequent pulses in a train (*Figure 5G*, green, *Figure 2—figure supplement 1A and B*) because its effect accumulates over multiple pulses. Importantly, this suppression grows with the integral of the stimulus over the duration of the filter lobe and hence with the pulse duration and duty cycle. In AN1$_M$, this leads to shorter pulse responses due to thresholding and saturation by the output nonlinearity (*Figure 2—figure supplement 1E*, *Figure 2—figure supplement 2*). Second and third, the adaptation in AN1$_M$ and LN3$_M$ accumulates during and across pulses and reduces these neuron's responses (*Figure 5G*, teal and red). This effect is again most prominent for pulse patterns with high duty cycles (long pulses, short pauses) since adaptation will be strongest during long pulses and recovery prevented during short pauses. Last, as discussed above, the inhibition from LN2$_M$ also suppresses responses in LN4$_M$ most strongly for stimuli with high duty cycle (*Figure 5G*, purple).

To examine how the different sources of suppression shape the model's tuning, we removed one or more of these computational steps: we set the inhibitory lobe of the AN1$_M$ filter to zero, we removed the adaptation from AN1$_M$ and LN3$_M$, and we removed the inhibition forwarded from LN2$_M$ to LN4$_M$ (*Figure 5G*). To accentuate the effects of these manipulations, we increased the delay of the LN5$_M$→LN3$_M$ connection, which led to a preference for longer pulse periods (50 ms) and for short duty cycles (0.25) in LN4$_M$ when all sources of suppression were present (*Figure 5H*, left). Consistent with the prediction that different sources of suppression in the network reduce responses for stimuli with high duty cycles, the network's preferred duty cycle tended to increase when suppression was removed (Figure 5H and I, *Figure 5—figure supplement 1*). Removing some sources of suppression tended to induce more sustained responses during a pulse train (*Figure 5G*) and to increase the preferred duty cycle from 0.25 to 0.50 (*Figure 5H and I*). Removing all four sources of suppression abolished period tuning and produced a preference for constant tone stimuli with a duty cycle of 1.0 (*Figure 5H*, right). Different sources of suppression sometimes interacted in unexpected ways. For instance, removing the inhibitory lobe and adaptation in AN1$_M$ decreased rather than increased duty cycle preference because AN1$_M$ produced stronger responses when adaptation was absent, which in turn induced stronger adaptation downstream in the network.

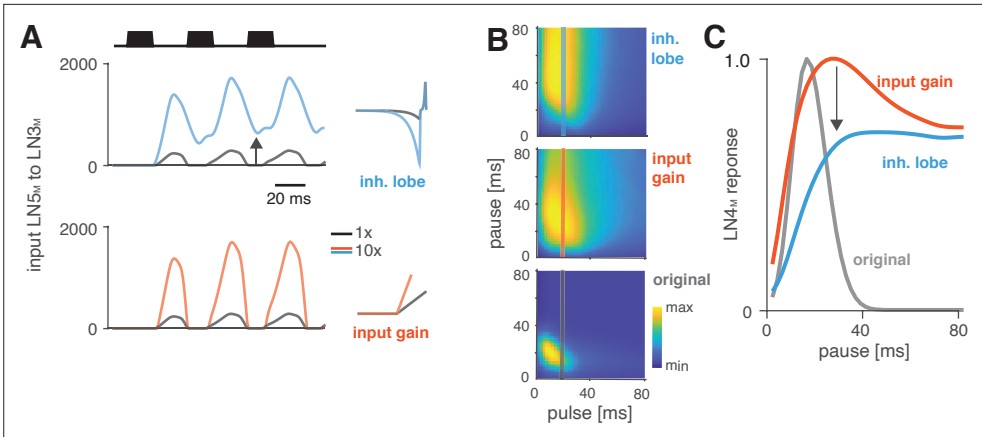

**Figure 6.** Changes in LN5$_M$ rebound dynamics induce a switch in response type. (**A**) Increasing the amplitude of the inhibitory lobe of the filter in LN5$_M$ that generates the rebound (middle, blue, 'inhibitory lobe') increases the rebound's amplitude and duration. By contrast, the gain of the input from LN5$_M$ to LN3$_M$ (bottom, red, 'input gain') scales the input from the LN5$_M$ rebound without prolonging it. Pictograms on the right show the parameters for the original model (black) and for a model with a 10-fold increase in the respective parameter value (blue and red). Traces show the rebound inputs from LN5$_M$ to LN3$_M$ for the pulse pattern shown on top (black traces, 20 ms pulse duration and 20 ms pause). (**B**) Response fields of LN4$_M$ for the original model (bottom), and for models with increased inhibitory lobe (top) or input gain (middle). Response magnitudes are color coded (see color bar, scaled to span the range of response magnitudes). Amplifying and prolonging the rebound by increasing the inhibitory lobe (top) produces pause tuning, while only amplifying the rebound via the input gain (bottom) retains period tuning (bottom). Vertical lines correspond to the stimuli for which pause tuning curves are shown in (**C**). (**C**) Pause tuning curves for LN4$_M$ at a pulse duration of 20 ms (see lines in **B**) reveal differential effects of the parameters on pause tuning. Amplifying and prolonging the rebound by increasing the inhibitory filter lobe (blue) produces high tolerance for pause duration, in this case high-pass tuning, which is required to obtain duration tuning. By contrast, only amplifying the rebound via the input gain (red) retains the preference for intermediate pauses characteristic for period tuning. The pulse trains had a duration of 600 ms and were interleaved by chirp pauses of 200 ms for all simulations to ensure that the stimuli in the response fields contained enough pulses even for long pulse durations and pauses.

The online version of this article includes the following figure supplement(s) for figure 6:

**Figure supplement 1.** Sustained rebound responses in LN5$_M$ induce stronger adaptation in LN3$_M$ for pulse patterns with short pauses.

---

Overall, these results identify mechanisms of response suppression by adaptation and inhibition as a cause for the network preferring small duty cycles (short pulses and long pauses). They demonstrate how specific implementation details of a recognition mechanism constrain phenotypic diversity, but also reveal how different model parameters can be used to create phenotypic diversity, like changing the preferred duty cycle (*Figure 5D,H, and I*). As a last step in our analysis, we examined a parameter that switches the preference type from period to duration tuning via changes in the rebound dynamics.

## Changes in rebound dynamics can switch the preference type by engaging suppression

Switches in preference type occur regularly even among closely related cricket species (*Bailey et al., 2017*; *Hennig, 2003*; *Hennig et al., 2016*), and we therefore looked for a model parameter that induced such a switch. We found that increasing the gain of the inhibitory lobe of the rebound filter in LN5$_M$ ('inhibitory lobe' in short) (*Figure 6A*, blue) switched the preference from period tuning (*Figure 6B*, bottom) to duration tuning (*Figure 6B*, top), characterized by high tolerance for pause and high selectivity for duration (*Figure 1B*). Increasing the inhibitory lobe parameter amplifies but also prolongs the rebound (*Figure 6A*, blue), and we examined which of these two changes creates the switch from period to duration tuning. Other model parameters – like the gain of the input from LN2$_M$ to LN5$_M$ or from LN5$_M$ to LN3$_M$ (*Figure 6A and B*, 'input gain,' red) or the LN5$_M$ output gain

– only amplify the rebound and retain period tuning. This indicates that the amplification of the rebound is insufficient, and that the prolongation of the rebound is necessary to cause the preference switch. In the original model, rebounds are short because the inhibition of $LN5_M$ triggered by $LN2_M$ activity always cuts off and suppresses rebound responses (*Figure 6A*, black, see also *Figure 2E and F*), and this happens even when the rebound is amplified (*Figure 6A*, red). By contrast, in the model with an increased inhibitory lobe of the $LN5_M$ rebound filter, the rebound persists during the $LN2_M$ inhibition (*Figure 6A*, blue). The prolonged rebound drives stronger adaptation downstream in $LN3_M$ (*Figure 6—figure supplement 1*), in particular for pulse patterns with short pauses, because shorter pauses prevent recovery from adaptation in $LN3_M$. This response suppression for short pauses abolishes the preference for intermediate pause durations necessary for period tuning and switches the preference type to duration tuning (*Figure 6C*). This last analysis highlights the dual role of suppression in shaping the recognition phenotype of the network: suppression constrains phenotypic diversity by reducing responses to patterns with long duty cycles (*Figure 4C and D*, *Figure 5H and I*), but it also contributes to phenotypic diversity by adjusting the network's preferred duty cycle (*Figure 5H, I*) or by switching the preference type (*Figure 6*).

## Discussion

How diversity in intraspecific communications systems is shaped by neural networks in the sender and in the receiver is an open question. Here, we asked whether the song recognition network in *G. bimaculatus* (*Schöneich et al., 2015*) has the potential to generate the diversity of song recognition phenotypes known from crickets. In particular, we tested whether the delay line and coincidence-detector network in *G. bimaculatus* can be considered a 'mother network' for recognizing the species-specific song pattern in different cricket species (*Figure 1*; *Hennig et al., 2014*). A model of the neuronal network reproduced the neurophysiological and behavioral data using simple, elementary computations like filtering, nonlinear transfer functions (nonlinearities), adaptation, and linear transmission with delays (*Figures 2 and 3*). Examining the model's responses over a wide range of parameter values revealed that the network can generate all types of song preferences known from different crickets and even other insects (*Figure 4A–E*). We then identified key parameters that either support or constrain the phenotypic diversity the network can produce, providing insight into how the network can evolve to become selective for different song parameters (*Figures 4F-H–6*).

### The delay line and coincidence-detector network can produce the full diversity of preferences for pulse duration and pause in crickets

Four principal preference types have been identified in crickets and other insects (*Figure 1B*): preference for pulse duration (*Deutsch et al., 2019*; *Gray et al., 2016*; *Hennig, 2003*), pulse pause (*Schul, 1998*), pulse period (*Hennig, 2003*; *Hennig, 2009*; *Rothbart and Hennig, 2012*; *Rothbart et al., 2012*), and duty cycle (*Hennig et al., 2016*). Variants of our network model produce all four of these preference types for the range of song parameter values relevant for crickets (*Figure 4A–E*).

While the network model analyzed here is derived from recordings in one species (*G. bimaculatus*), the delay line and coincidence-detector network is likely shared within the closely related cricket species. The phylogenetic position of *G. bimaculatus* close to the base of the phylogenetic tree from which many other species emerged is consistent with this idea (*Gray and Cade, 2000*). Our finding that this network can produce all known preferences for pulse and pause supports this idea and suggests that it forms a common substrate – a 'mother network' – for the diversity of song recognition phenotypes in crickets. How can the 'mother network' hypothesis be tested? Behavioral tests can provide insight into whether other species use the coincidence-detection algorithm found in *G. bimaculatus* (*Hedwig and Sarmiento-Ponce, 2017*). These experiments can, for instance, test the prediction that the duration of the last pulse in a chirp only weakly impacts network responses. Species that violate this prediction are unlikely to recognize song by the same coincidence mechanism. However, the 'mother network' hypothesis does not imply that all crickets implement a coincidence-detection algorithm, just that they reuse the same neurons with largely conserved response properties. In fact, our analyses have shown that coincidence detection can be circumvented through changes in key parameters to produce a different preference type (*Figure 6*, *Figure 4—figure supplement 1*). That is why further electrophysiological experiments in *G. bimaculatus* are crucial to reveal the precise

biophysical mechanisms that tune the network and ultimately link changes in gene expression, for instance, of specific ion channels, to changes in network tuning. Importantly, these experiments need to be extended to other species by identifying and characterizing homologues of the neurons in the network. Recordings in other species are challenging but feasible since homologous neurons are expected to be found in similar locations in the brain. Our model produces testable predictions based on the known behavioral tuning for how key properties of these neurons may look like in any given species (see below).

Future studies will also show whether the network can explain more complex preference functions known from some crickets and also other insects. For instance, preference types that betray resonant cellular or network properties are known from katydids (*Bush and Schul, 2005*; *Webb et al., 2007*), and we find that the network can produce these multi-peaked response fields (*Figure 4—figure supplement 2D*), but it remains to be seen whether similar preference types exist in crickets. Several species of crickets produce complex songs that are composed of multiple types of pulse trains, and it is unclear whether the current network can reproduce the known behavioral preference for such complex songs (*Bailey et al., 2017*; *Cros and Hedwig, 2014*; *Hennig and Weber, 1997*).

In addition, we have not yet explored the network's ability to reproduce the behavioral selectivity for parameters on the longer timescale of chirps (*Figure 1A*; *Grobe et al., 2012*; *Blankers et al., 2016*; *Hennig et al., 2016*). It is likely that the network can explain some properties of the selectivity for chirp known from crickets. For instance, that a minimum of two pulses is required to produce coincidence in the network could at least partly explain the existence of a minimal chirp duration for *G. bimaculatus* (*Grobe et al., 2012*). Likewise, suppression in the network reduces responses to long chirps, which could explain the reduced behavioral preference for long chirps. However, the current electrophysiological data do not sufficiently constrain responses at these long timescales and studies are needed to address this issue more comprehensively.

Lastly, our model does not address the behaviourally well-documented inter-individual variability in phonotaxis behaviors (*Grobe et al., 2012*; *Meckenhäuser et al., 2014*), which likely arises at multiple levels: At the level of song pattern recognition (inter-individual differences in the network parameters) at the level of phonotaxis behavior (biases and noise in localizing the sound) at the motivational level (low or high motivation leads to more or less selective responses) and at the motor level (variability from motor noise). Identifying the contribution of these different levels is challenging since the full characterization of the behavioral phenotype in terms of the response fields cannot be obtained reliably at the individual level - the stimulus space is too large. Therefore our model of song pattern does not explicitly consider the inder-individual variability but is meant to represent he behavior of an average female.

## How to tune a pulse pattern detector?

Our sensitivity analysis of the model identified three classes of parameters that define the model's tuning (*Figure 4H*). First, parameters that control the relative timing of inputs to the coincidence-detector LN3$_M$ set the network's preferred pulse period. These include input delays in all upstream neurons (*Figure 5A and B*) but also passive and active membrane properties that delay the rebound responses in LN5$_M$ (*Figure 5—figure supplement 1*). Second, parameters that lead to a stronger and more sustained rebound in LN5$_M$ can shift the preference from pulse period to pulse duration tuning (*Figure 6*). Lastly, sources of response suppression, like inhibition or adaptation, reduce responses to long pulses and high duty cycles (*Figure 5G–I*). These three classes of parameters account for changes within and transitions across the principal types of song preference in crickets. The model thus provides testable hypotheses for how response properties in the neuronal network may have evolved to compute the preference functions of different species. For instance, species that prefer different pulse periods than *G. bimaculatus* like *Teleogryllus leo* (*Rothbart and Hennig, 2012*), *Gryllus locorojo* (*Rothbart et al., 2012*), *Gryllus firmus* (*Gray et al., 2016*), or *Teleogryllus oceanicus* (*Hennig, 2003*) could differ in the delays of inputs to LN3 (*Figure 5B*). Duty cycle-tuned species like *G. lineaticeps* or G15 (*Hennig et al., 2016*) may exhibit weaker suppression throughout the network, for instance, reduced adaptation in LN3 (*Figure 5G–I*). Lastly, species with duration tuning such as *T. commodus* (*Hennig, 2003*) or G13 (*Gray et al., 2016*) could exhibit longer and stronger rebound responses in LN5 (*Figure 6*).

How can these changes be implemented in a biological network? Although our phenomenological model is independent of a specific biophysical implementation, all model components have straightforward biophysical correlates. We can therefore propose biophysical parameters that tune specific aspects of the network in a given implementation. To illustrate this point, we will briefly provide examples of how the four elementary computations of the model – filtering, adaptation, nonlinear transfer functions (nonlinearities), and linear transmission with a delay – can be implemented and tuned. First, filters are shaped by active and passive properties of the membrane: individual filter lobes act as low-pass filters that dampen responses to fast inputs and arise from integrating properties of the passive membrane like capacitive currents (*Dewell and Gabbiani, 2019*; *Azevedo and Wilson, 2017*). Increasing the membrane capacitance therefore leads to stronger low-pass filtering. An added negative (inhibitory) lobe makes the stimulus transformation differentiating and can arise from conductances that hyperpolarize the membrane, like potassium or chloride channels (*Nagel and Wilson, 2011*; *Slee et al., 2005*; *Lundstrom et al., 2008*). Increasing the potassium conductance in *Drosophila* olfactory receptor neurons makes their responses more differentiating (*Nagel and Wilson, 2011*), while reducing the conductance of delayed-rectifier potassium channels in the auditory brainstem makes responses less differentiating (*Slee et al., 2005*; *Lundstrom et al., 2008*). The filter in LN5$_M$ that produces the post-inhibitory rebound arises from hyperpolarization-activated cation currents like I$_h$ (mediated by HCN non-selective cation channels) and I$_t$ (mediated by T-type calcium channels), which control the PIR's amplitude and latency, respectively (*Pape, 1996*; *Engbers et al., 2011*). Second, adaptation is implemented in the model either via inhibitory filter lobes or via divisive normalization. Biophysically, adaptation can arise from synaptic depression (*Tsodyks et al., 1998*; *Fortune and Rose, 2001*), or from subthreshold or spike-frequency adaptation (*Farkhooi et al., 2013*; *Nagel and Wilson, 2011*; *Benda and Herz, 2003*; *Benda and Hennig, 2008*). Spike-frequency adaptation can arise from inactivating sodium currents, voltage-gated potassium currents (M-type currents), or calcium-gated potassium currents (AHP currents) (*Slee et al., 2005*; *Heidenreich et al., 2011*). Increasing the expression level of these channels controls the strength of adaptation, while the kinetics specific to each channel type control adaptation speed. AHP currents can last seconds if spiking leads to a long-lasting increase in the intracellular calcium concentration, giving rise to long inhibitory filter lobes or adaptation time constants in the model (*Cordoba-Rodriguez et al., 1999*). Third, nonlinearities translate the integrated synaptic input to firing output. The nonlinearity's threshold is governed by the density of sodium channels at the spike-initiating zone while the steepness and saturation of the nonlinearity depend on the inactivation kinetics of sodium channels or the spiking dynamics controlled by the Na/K ratio (*Prescott et al., 2008*; *Lundstrom et al., 2008*). Lastly, transmission delays arise from axonal conduction and synaptic delays but also other mechanisms, like low-pass filtering of the membrane voltage at the pre- and postsynapse (*Creutzig et al., 2010*; *Zhou et al., 2019*), latencies arising from integration of inputs to the spiking threshold, or from spike generation (*Izhikevich, 2004*; *Figure 5—figure supplement 1*). Synaptic weights are set by the number of synaptic boutons between two neurons (Peter's rule, see *Rees et al., 2017*) and the amount of neurotransmitter that can be released at the presynapse (vesicle number and loading) and absorbed at the postsynapse (number of transmitter receptors). These examples demonstrate that our phenomenological model has a straightforward, physiologically plausible implementation and can propose experimentally testable hypotheses for transitions between types of behavioral preferences.

The three computations that define model tuning – response suppression, post-inhibitory rebounds, and coincidence detection – occur across species and modalities. Delay lines are prominent in binaural spatial processing (*Schnupp and Carr, 2009*) but have also been implicated in visual motion detection (*Borst and Helmstaedter, 2015*) or pulse duration selectivity in vertebrate auditory systems (*Aubie et al., 2012*; *Buonomano, 2000*). Suppression is known to act as a high-pass filter for pulse repetition rates (*Baker and Carlson, 2014*; *Benda and Herz, 2003*; *Fortune and Rose, 2001*) that in our case biases the network towards responding to rapidly changing patterns, like those with short pulses (*Figure 4C*). Finally post-inhibitory rebounds have been implicated in temporal processing in different species like honeybees (*Ai et al., 2018*), fish (*Large and Crawford, 2002*), frogs (*Rose et al., 2015*), or mammals (*Felix et al., 2011*; *Kopp-Scheinpflug et al., 2018*). The computations found to control the song preference in *G. bimaculatus* could therefore also govern pattern recognition in other acoustically communicating animals. For instance, different bird species produce songs with silent gaps of species-specific durations between syllables and auditory neurons in the bird's brain are sensitive to

these gaps (*Araki et al., 2016*). Our analyses suggest that the gap preference can be shifted from longer gaps (low duty cycle: Java sparrow, Bengalese finch) to shorter gaps (high duty cycle, starling, zebra finch) by reducing suppression or adaptation in the network. This could be implemented, for example, by reducing postsynaptic GABA receptors or by lowering the expression levels of voltage-gated potassium channels.

We here focused on modifying the magnitude of parameters, corresponding, for instance, to the expression levels of neurotransmitters or ion channels. Neuronal networks, however, can also evolve to produce novel phenotypes by changing their topology, through a recruitment of novel neurons, a gain or loss of synapses, or switches in synapse valence from excitatory to inhibitory as has been shown in motor networks in *Caenorhabditis elegans* (*Hong et al., 2019*) and snails (*Katz, 2011*). In addition, we have not considered neuromodulators, which can rapidly alter network tuning (*Bargmann, 2012*; *Marder, 2012*; *Marder et al., 2014*), and which likely play a functional role in the phonotactic response (*Poulet and Hedwig, 2005*).

## Algorithmic details specify constraints

Previous studies revealed that Gabor filters can produce the full diversity of song preference functions found in insects (*Clemens and Hennig, 2013*; *Clemens and Ronacher, 2013*; *Hennig et al., 2014*). However, the computation giving rise to Gabor filters can be implemented with multiple algorithms, each subject to specific constraints. For instance, the period tuning found in *G. bimaculatus* can be produced by the now known delay line and coincidence detection mechanism (*Schöneich et al., 2015*), but also by the interplay between precisely timed excitation and inhibition (*Aubie et al., 2012*; *Rau et al., 2015*), by cell-intrinsic properties like resonant conductances (*Azevedo and Wilson, 2017*; *Rau et al., 2015*), or by a combination of synaptic depression and facilitation (*Fortune and Rose, 2001*). By considering the implementation of the pattern recognition algorithm in a particular species, we revealed a bias in the diversity of phenotypes that this specific implementation can produce: several sources of suppression induce a bias towards preference for low duty cycle stimuli (*Figures 4B–D and 5G–I*). This highlights the importance of studying nervous system function and evolution beyond the computational level at the level of algorithms and implementations (*Marr, 1982*).

## Functional tradeoffs limit behavioral diversity

The low duty cycle bias present in the recognition mechanism of *G. bimaculatus* has several implications for the evolution of song preference in crickets and elsewhere: perceptual biases that have evolved in contexts like food or predator detection are known to shape sexual selection (*Guilford and Dawkins, 1993*; *Ter Hofstede et al., 2015*; *Phelps and Ryan, 1998*; *Ryan and Cummings, 2013*). In the case of song recognition in crickets, suppression (adaptation, inhibition, onset accentuation, *Figure 5G*) reduces neuronal responses to long-lasting tones and likely evolved to save metabolic energy (*Niven, 2016*) or to make song recognition more robust to changes in overall song intensity (*Benda and Hennig, 2008*; *Hildebrandt et al., 2011*; *Schöneich et al., 2015*). As a side effect, adaptation now biases the song recognition mechanism towards preferring pulse trains with low duty cycles (*Figure 5H and I*), which is consistent with the apparent absence of pause tuning in crickets (*Hennig et al., 2014*). Interestingly, pause tuning is known from katydids (*Schul, 1998*), suggesting that their song recognition system is not subject to the low duty cycle bias. Katydids may have avoided the low bias either by using a delay line and coincidence detection network like that found in *G. bimaculatus* but with weaker suppression (*Figure 5G–I*) or by using a different network design that is subject to different constraints (*Bush and Schul, 2005*). Thus, computations that increase energy efficiency and robustness can constrain the phenotypic diversity of a whole species group.

## From evolutionary pattern to process

How can a diversity of neural networks evolve to drive the diversification of species-specific communication signals? Our modeling study of the song recognition network in the cricket brain provides first evidence that the underlying neuronal network is computationally flexible: by adapting physiological parameters, the network can produce all preference types described in crickets (*Figure 4B–E*). The computational flexibility of the recognition mechanism may explain the species richness as well as the speed of evolution in a particular taxon like crickets (*Alexander, 1962*; *Blankers et al., 2015*; *Desutter-Grandcolas and Robillard, 2003*; *Oh and Shaw, 2013*; *Otte, 1992*): female preferences drift around

with little constraint in signal space, maybe pushed by abiotic (environmental noise selects against preferences for very short pauses) and biotic factors (avoid overlap with heterospecifics, *Amezquita et al., 2011*). The male song evolution follows changes in the female's preference since only males that sing attractive song will reproduce. In this scenario, a female network that has the capacity to produce many different preference types supports the divergence of the communication system. However, this co-evolution of song preference and song structure requires male song production networks to be as flexible as the female song recognition networks (*Jacob and Hedwig, 2020*; *Schöneich, 2020*). There is also accumulating evidence for genetic coupling between the networks that produce and recognize the song pattern, which may ensure that sender and receiver stay tuned during evolution of song pattern (*Xu and Shaw, 2019*; *Schöneich, 2020*; *Xu and Shaw, 2021*).

Computational flexibility also needs evolvability – the ability to generate specific and robust phenotypes during evolution (*Blankers et al., 2015*). The 'mechanistic degeneracy' of computation in biological neural networks – the fact that specific network outputs can be obtained by tuning disparate parameters – is thought to be a prerequisite for evolvability (*Wagner, 2011*, see also *Hasson et al., 2020*; *Leonardo, 2005*; *Schneider et al., 2021*). The degeneracy concept has been demonstrated extensively for motor networks (*Gutierrez et al., 2013*; *Prinz et al., 2004*), and here we also found evidence of degeneracy in the cricket song recognition network. For instance, the preferred pulse period (*Figure 5A and B*, *Figure 5—figure supplement 1*) or the pulse duty cycle (*Figure 5G–I*) can be set by multiple parameters. The hypothesis that high-dimensional and degenerate biological parameter spaces are more evolvable could be tested by assessing the computational flexibility of a minimal model of the network that produces the observed behavior with as few parameters as possible.

If a neural network is flexible, how can it maintain a robust phenotype within a species? We find that many parameters have only weak impacts on the network's preference, leading to a relatively large fitness-neutral parameter set and a robust phenotype (*Prinz et al., 2004*). For instance, we observed that the rebound amplitude in LN5 tends to be variable across electrophysiological recordings in different individuals but that the network model's output is relatively robust to these changes (*Figure 2B*, *Figure 2—figure supplement 4*). Other parameters crucially affect the recognition phenotype and support evolvability. These fitness-critical parameters likely drive changes in the recognition phenotype during speciation events to ensure species discrimination (*Amezquita et al., 2011*; *Schmidt et al., 2011*). Future studies will employ novel computational methods for characterizing the high-dimensional parameter space (*Bittner et al., 2021*; *Gonçalves et al., 2020*) to more comprehensively characterize the fitness-neutral parameter set leading to interindividual variability and to provide insight into the robustness and evolvability of pattern recognition networks.

## Materials and methods
### Electrophysiological data
The data used for fitting the model come from intracellular recordings with sharp microelectrodes of AN1, LN2, LN3, LN4, and LN5 and are published in *Kostarakos and Hedwig, 2012* and *Schöneich et al., 2015*. They include (1) 12 stimuli with a pulse duty cycle of ~0.5 and periods ranging from 10 to 98 ms (*Figure 2B*), (2) 10 stimuli with a pulse period of 40 ms and duty cycles ranging between 0.1 and 1.0 (*Figure 2C*), and (3) 12 stimuli with a pulse duration of 20 ms and pauses varying between 6 and 90 ms (*Figure 2D*). During the electrophysiological experiments, each pulse train was presented interleaved by a chirp pause of 230 ms.

### Quantification of tuning in the recordings and the model responses
In the recordings, spikes were detected using custom routines that found peaks in the voltage traces robust to changes in baseline. The accuracy of spike detection was checked by visual inspection of the voltage traces. From the spike times, average firing rates were calculated by dividing the number of spikes produced during each chirp (pulse train and chirp pause) by the chirp period (*Figure 2B–D*). For the non-spiking LN5, the response corresponds to the voltage of the rebound response. Voltage rates were obtained by first calculating a baseline voltage as the average voltage in the 25 ms preceding a given pulse train, and then integrating the supra-threshold components of the voltage. This integral voltage was then divided by the chirp period to get a rate. Note that the tuning curves for LN5 are

not very accurate because the rebound responses have a low amplitude and the baseline voltage fluctuated considerably during recordings of LN5.

For the model, tuning curves and responses fields were calculated as for the recordings – by integrating the firing rate outputs of the spiking neurons in the model or the rebound voltage for the non-spiking LN5 and dividing the resulting values by the chirp period.

## Model inputs and simulation

We built a rate-based, phenomenological model of the song recognition network in *G. bimaculatus*. Sound inputs were presented as the time-varying amplitude of pulse trains with the specified pulse and chirp structure. Model responses were simulated with a temporal resolution of 1 kHz and tested as if pulse trains and chirp pauses were repeated endlessly.

## Elementary computations

The model was built from four elementary neural computations. In the following, we will first define the elementary computations and then describe how they were combined to model each neuron in the network. Neurons in the model are referred to with a subscript $M$.

### Filtering

Filtering was implemented via $y(t) = \int_\tau h(\tau)x(t - \tau)$, where $x(t)$ and $y(t)$ are the stimulus and response at time $t$, respectively, and $\tau$ is a temporal delay. The causal filter $h(\tau)$ was constructed from discrete Gaussian and exponential kernels. A discrete Gaussian kernel was defined as $h_g = e^{-(\tau - N/2)^2/(4\sigma^2)}$ with support $N$, where $0 \leq \tau \leq N$, and width $\alpha$ such that $\sigma = (N - 1)/(2\alpha)$. Exponential kernels were defined as $h_e = e^{-\tau/\gamma}/\gamma$, with support $N$, where $0 \leq \tau \leq N$, and decay $\gamma$. Note that throughout the article we use 'filter duration' as a non-technical synonym for 'support,' which specifies the time over which the filter is defined. Gaussian (*Figure 2—figure supplement 1A*) and exponential filters are low-pass or integrating filters, which smooth the stimulus (*Figure 2—figure supplement 1B*). The filters were not normalized. Bandpass filters were implemented as biphasic filters by either differentiating a Gaussian filter $\Delta h(\tau) = h(\tau) - h(\tau - 1)$ (*Figure 2—figure supplement 1A*) or by subtracting two Gaussian and/or exponential filters to form the positive (excitatory) and negative (inhibitory) filter lobes. Parameterizing biphasic filters as combinations of Gaussians and/or exponentials provides greater and more direct control over the timing and the gain of the filters' excitatory and inhibitory lobes. Differentiating filters respond most strongly to changes in the stimulus, for instance, to the onsets or offsets of pulses (*Figure 2—figure supplement 1B*). The support parameter was initialized to be sufficiently long and was typically fixed during model fitting (*Table 1*).

### Nonlinear transfer functions

Nonlinear transfer functions (nonlinearities) transform inputs to implement thresholding or saturation. The model employs two types of nonlinearities: a rectifying nonlinearity thresholds the input $x$ at the threshold value $x_0$ and scales it with a gain $\beta$:

$$y = \begin{cases} x \cdot \beta & \text{if } x > 0 \\ 0 & \text{otherwise} \end{cases}$$

(*Figure 2—figure supplement 1D*). In many cases, the threshold parameter was used to constrain signals to be purely positive or negative, and it was therefore often fixed to 0. A sigmoidal nonlinearity combines a soft threshold with saturation: $y = y_0 + y_{max}/(1 + e^{-ax-b})$ with gain $a$, shift $b$, and minimal and maximal output $y_0$ and $y_{max}$, respectively (*Figure 2—figure supplement 1E*).

### Linear transmission with delay

Neuronal connections transmit neural activity linearly with delay $\Delta$ and gain $\alpha$. Multiple inputs to a neuron are added. The gain $\alpha$ is negative for inhibitory and positive for excitatory synapses and controls the input strength. The delay parameter $\Delta$ corresponds to the delay that needs to be added to the input of a neuron to reproduce the timing of the output of that neuron. It includes axonal conduction or synaptic transmission delays but also other delay mechanisms, like low-pass filtering

of the membrane voltage at the pre- and postsynapse (*Creutzig et al., 2010*; *Zhou et al., 2019*) or latencies arising from integration of inputs to the spiking threshold or from spike generation.

## Adaptation

Adaptation in $AN1_M$ and $LN2_M$ is implemented using differentiating filters (*Figure 2—figure supplement 1A and B*). Such filters produce adaptation via their broad inhibitory lobe, which suppresses subsequent responses (*Nagel and Wilson, 2011*). For $AN1_M$ and $LN3_M$, adaptation effects were better captured using divisive normalization (*Figure 2—figure supplement 1F*). The input to the adaptation stage, $x$, is low-pass filtered using an exponential filter to generate an adaptation signal $x_{ada}$ that divides the input: $y = x/(x_0 + w \cdot x_{ada})$. Here, $x_0$ and $w$ control the strength of adaptation while the width of the exponential filter, $\gamma$, controls the timescale of adaptation. Both implementations of adaptation – differentiating filters and divisive normalization – can produce similar adaptation time courses. However, since adaptation with differentiating filters acts subtractively, the negative response components accumulate over time to a negative filter output (*Figure 2—figure supplement 1F*). For modeling adaptation on longer timescales, this can completely suppress responses to subsequent stimuli. This complete suppression does not occur with divisive normalization since it only scales the response.

## Model neurons

The elementary computations described above were combined to reproduce the response of each neuron in the network – the firing rate patterns in the case of spiking neurons and the pattern of the rebound responses in the non-spiking $LN5_M$ (*Figure 2E*). All filters and nonlinearities are plotted in *Figure 2A*, and all parameter names and values are listed in *Table 1* and graphically defined in *Figure 2—figure supplement 1* and *Figure 2—figure supplement 2*. The model was implemented in MATLAB. Code for running the model is available at https://github.com/janclemenslab/cricketnet, (*Clemens, 2021* copy archived at swh:1:rev:73ff788143c6afa3adbbb3e0f61f600a13052352).

### $AN1_M$ (relay)

AN1 is the input neuron of the network. It faithfully copies the pulse structure and adapts weakly (*Figure 2E*). These response properties were reproduced using a differentiating linear filter (*Figure 2—figure supplement 1*): the filter's narrow and strong excitatory lobe captures the fast and faithful stimulus responses, while the broad inhibitory lobe captures the delayed suppressive effect of past stimulus epochs. $AN1_M$'s filter was generated by concatenating two Gaussians as the excitatory and inhibitory lobes. $AN1_M$ has a sigmoidal nonlinearity that saturates responses to longer pulses. To capture the decrease of the response across pulses, we additionally included a divisive normalization stage.

### $LN2_M$ (inhibition)

LN2 responds reliably to individual pulses similar to AN1, but exhibits stronger adaptation during the pulse. This was reproduced using a differentiating filter, composed of a truncating Gaussian function as the excitatory lobe and an exponential function as the inhibitory lobe. A rectifying nonlinearity restricts outputs to be positive.

### $LN5_M$ (post-inhibitory rebound)

LN5 is a non-spiking neuron. It receives inhibitory input coupled to the spike activity of LN2 and generates a rebound depolarization when the inhibition terminates (*Figure 2E*). The inhibitory input from LN2 over a pulse saturates in LN5, which is modeled using a differentiating input filter followed by a rectifying nonlinearity that restricts the inputs to be negative. The rebound is then produced using a differentiating filter with a narrow excitatory lobe, which further low-pass filters the inhibitory input, and a broad inhibitory lobe, which generates the broad positive voltage deflections at the offset of negative voltage deviations. See *Figure 2—figure supplement 1C* for an illustration of the principle by which the rebound is produced in the model. The filter was generated by concatenating two exponential filters as negative and excitatory lobes.

### LN3 (coincidence detection)

LN3 responds strongly when two excitatory synaptic inputs arrive at the same time: a short-latency input from AN1 and a delayed input from LN5. In the neurophysiological data, the input from AN1 alone is often sufficient to drive spiking (e.g., to the first pulse of a train), suggesting that LN3 does not require two coincident inputs to spike. We therefore define $LN3_M$ as a linear adder (with a threshold) that receives stronger input from $AN1_M$ than from $LN5_M$. From the $LN5_M$ responses, only the positive components corresponding to the rebound were passed on as the input to $LN3_M$, which was separated from the negative components of the $LN5_M$ responses using a rectifying nonlinearity (*Figure 2—figure supplement 1D*). The sum of the two inputs is thresholded using a rectifying nonlinearity before being passed into an adaptation stage with divisive normalization. The adaptation after summation reduces $LN3_M$ firing over a pulse train, and a final rectifying nonlinearity ensures positive firing rates, by cutting of all negative responses that can occur in the model, for instance, due to filtering (*Figure 2—figure supplement 1B*).

### LN4 (feature detection)

Finally, $LN4_M$ adds excitatory input from $LN3_M$ and inhibitory input from $LN2_M$, which sharpens its selectivity for pulse patterns as compared to $LN3_M$. A rectifying nonlinearity restricts the output firing rate to be positive.

## Prediction of phonotactic behavior from network responses

For regular pulse trains, the firing rate of LN4 (number of spikes per chirp/chirp duration) predicts phonotactic behavior very well (*Schöneich et al., 2015*; *Figure 1E*). We therefore used the firing rate of $LN4_M$ to predict behavioral responses from network output.

## Model fitting

Model parameters were optimized in two phases. First, the model parameters and structure were initialized by hand. This initialization established which computational steps were necessary to reproduce the key response features of the neurons in the network – their response dynamics and tuning – and determined initial parameter values. In the second phase, we used a genetic algorithm to tune the network parameters to optimally fit the data; see *Clemens et al., 2014*. To simplify fitting, we exploited the feed-forward topology of the network: instead of fitting all parameters simultaneously, we adopted a stepwise procedure, in which the model neurons were fitted in order of their position in the network. We started with fitting the parameters of the first neuron in the network, $AN1_M$, and reproducing the firing rate traces of $AN1_M$. In a seconds step, we held the parameters of $AN1_M$ constant and fitted parameters of $LN2_M$. We then held the parameters of $AN1_M$ and $LN2_M$ constant and fitted the temporal dynamics of the rebound response in $LN5_M$ but not the inhibitory component, since the precise magnitude and dynamics of inhibitory components were highly variable across recordings and since the rebound constitutes the effective output of LN5. We then fitted the parameters of $LN3_M$ and finally of $LN4_M$, again holding those of the upstream neurons constant. For these last two neurons, firing was very sparse and irregularly timed across trials, which complicated fitting. We therefore used information from the firing rate dynamics and from the tuning curves for optimizing the parameters of $LN3_M$ and $LN4_M$ by using mixed error functions: $E_{MSE}^{traces} + w \cdot E_{MAE}^{tuning}$ where $E_{MSE}^{traces}$ is the mean square error between the predicted and actual traces, $E_{MAE}^{tuning}$ is the mean absolute error between the predicted and the actual tuning curves, and $w$ was set to 0.1 for $LN3_M$ and to 1.0 for the even sparser activity of $LN4_M$. For the other neurons in the network ($LN2_M$, $LN3_M$, $LN5_M$), only E was used (equivalent to w = 0).

$$E_{MSE}^{traces}$$

The initial parameters for fitting at each step were drawn from an exponential distribution ranging between 0.1 and 10 around the hand-fitted parameters. The exponential distribution was chosen to have an even number of parameters above and below the hand-fitted values, that is, a uniform distribution after log scaling. While the fitting algorithm is stochastic by nature, it repeatedly found similar optimal parameter sets on independent initializations (not shown).

The model had a total of 55 parameters. Of these, nine parameters (marked by '*' in *Table 1*) were fixed to the initialized values during fitting to facilitate convergence. For instance, the gain of the input to a neuron is equally set by the output gain of the presynaptic neuron's nonlinearity, $\beta$, or by the synaptic input gain $\alpha$, and therefore only one of these parameters needs to be optimized. Or the support parameter of an exponential filter sets an upper bound for the timescale of that filter, which in turn is given by the filter's time constant. In that case, the support was typically set to a sufficiently high value and fixed. Lastly, some of the thresholds were fixed to zero since their sole role was to restrict firing rates to be positive.

## Evaluation of model performance

After the fitting procedure, the model performance was evaluated based on the tuning curves and the response traces. The fit for the tuning curves is given by $1 - E_{RMSE}(a_n, b_n)$ (*Figure 2H*), where $E_{RMSE}$ is the root-mean-squared error and $a_n$ and $b_n$ are the tuning curves from the experimental data and the model, each normalized by the maximum of the tuning curve from the data: $a_n = a/\max a$, $b_n = b/\max a$. This measure of fit is sensitive to the addition of a constant offset and to a mismatch in the scale of responses between the model and the data, but is independent of the overall scale of the responses in the data. The fit for the response traces is given by the squared Pearson correlation coefficient:

$$r^2 = \left( \frac{\sum_{i=1}^{n}(x_i - \bar{x})(y_i - \bar{y})}{\sqrt{\sum_{i=1}^{n}(x_i - \bar{x})^2}\sqrt{\sum_{i=1}^{n}(y_i - \bar{y})^2}} \right)^2$$

where $x$ and $y$ are the response traces from the model and the data, respectively, and $\mu$ and $\sigma$ are the mean and the standard deviation. The $r^2$ was calculated for different timescales (*Figure 2G*) by low-pass filtering both the prediction and the response with rectangular windows with durations ranging between 1 and 25 ms before computing the $r^2$. At short timescales, this measure is sensitive to fine details in the firing rate dynamics, while at longer timescales, the measure reflects the match in coarse features of the firing dynamics, for example, the spike counts per pulse.

## Generation of model variants for the analysis of the parameter space

To determine the range of preference types the network can produce (*Figure 4A–E*), we generated models with random parameter sets taken from a range around that obtained from the fit to *G. bimaculatus*. For that, we chose 45 of the 55 model parameters. The 10 parameters fixed for this analysis (marked by '†' in *Table 1*) had also been fixed for fitting (see above), but we allowed more of the parameters of AN1's filter and output nonlinearity to change. All 45 parameters except the synaptic delay parameters were taken from a hypercube spanning the range between 1/10 and 10-fold around the original parameter set, spaced logarithmically such that the fraction of random parameter values below and above the original value was similar. Delay parameters in the model (see *Table 1*) were allowed to range uniformly between 1 and 21 ms, irrespective of the original parameter value. The delay parameters correspond to the delay added to the inputs to a neuron required to produce the desired timing of the output of that neuron and include axonal conduction and synaptic transmission delays, delays induced by low-pass filtering at the pre- and postsynapse, and delays from the integration of inputs to the spiking threshold and from spike generation. To ensure uniform sampling from the 45-dimensional parameter space, we used a quasi-random sampling scheme based on the Sobol set. We used the sobolset function in MATLAB with the following parameters: skip 1e3, leap 1e2, scramble `MatousekAffineOwen`. Using this approach, we generated 5 million different model variants. Of these, 9% (450,000 models) were responsive and selective, that is, they responded to at least one pulse train pattern and did not produce the same response for all patterns tested. Initial tests with extended parameter ranges yielded qualitatively similar phenotypic variability but produced many more unresponsive or unselective models.

## Sensitivity analysis

For the sensitivity analysis (*Figure 4F–H*, *Figure 4—figure supplement 1*), we changed either single parameters or systematically varied pairs of parameters on a grid. We used the same set of 45 parameters as in the analysis of the parameter space above (see *Table 1*). Single-parameter sweeps were generated as 21 logarithmically spaced values between 1/100 and 100-fold around the original value,

except for delay parameters, which were generated as 21 values ranging between 1 and 41 ms. While these parameter ranges may appear rather large, in particular for the input delay parameters, they are chosen to facilitate the detection of key parameters controlling the network's tuning. Their specific and meaningful effects on the model tuning were confirmed and further analyzed within the more restricted parameter range also chosen for the analysis of the network's phenotypic flexibility above.

For each model, we calculated the response fields for pulse durations and pauses between 1 and 80 ms (2 ms spacing, 1600 stimuli per response field), with a pulse train duration of 600 ms and a chirp pause of 200 ms. To quantify the model's sensitivity to changes in each parameter, we first calculated the correlation distance (1 – Pearson's correlation coefficient) between the response fields from the original model and each modified model in the parameter sweep (*Figure 4F*). By using the correlation coefficient, our sensitivity analysis is robust to trivial changes in the response field like scaling or the addition of a constant to all responses. The average correlation distance over a parameter sweep is then taken as a measure of how much changing a parameter affects the model output.

For some parameters, models produced constant output (e.g., all zeros) over most of the parameter sweep with only one or two step-like changes in the response fields, leading to artificially high sensitivity values. We found that we could reliably exclude such parameters by calculating the median difference in the correlation distance between consecutive parameter values over the parameter sweep and requiring this quantity to be larger than 0.005:

$$median_{-N \leq n < N} |D(R(0), R(n)) - D(R(0), R(n+1))| > 0.005$$

where $|\ldots|$ is the absolute value, and $D(R(0), R(n))$ denotes the correlation distance (1 – Pearson's correlation coefficient) between the response field from the original model, $R(0)$, and from a model with the modified parameter, $R(n)$. This approach also excluded parameters whose change produced largely untuned models over the sweep. We ensured that none of our results crucially depended on these criteria. For instance, even without the above constraint, the parameters of $LN5_M$ and $LN3_M$ were still among the top ranked.

For the sensitivity analysis over parameter pairs (*Figure 4—figure supplement 1*), value grids were generated using the same value ranges as for single parameters for each parameter in the pair, resulting in 21 × 21 = 441 model variants for each of the 1035 unique, unordered parameter pairs.

## Characterization of response fields

The response fields known in crickets typically have a roughly ellipsoid shape with a single peak. To assess the extent by which the response fields produced by randomizing the model parameters match these properties, we determined the fraction of fields with a single dominant peak (vs. multiple dominant peaks) and the match of the response fields with a best-fitting ellipse. Dominant peaks were defined as local maxima in the response field exceeding 0.5 of the global maximum value and separated by a trough that is smaller than 0.75 of the value of the lower of the two peaks. This measure is robust to the existence of either small or poorly separated local maxima. Ellipses were fitted to the binarized fields to be robust to the steps in the response fields by the stimulus structure (*Figure 3—figure supplement 1*). Binarization was performed by thresholding each response field at 50% of its maximum value. We then fitted a two-dimensional Gaussian to the superthreshold coordinate values using MATLAB's `fitgmdist` and created a best-fitted prediction by setting all pixels inside the 99.9% probability threshold to 1.0. Results are robust to the specific choice of this value. The match between the binarized response field and the binarized ellipsoid was estimated via the Jaccard similarity, $S_J$, between the two binary images. The Jaccard similarity corresponds to the fraction of non-zero pixels that differ between the binarized response fields and the binarized ellipsoid or the intersection over union: $S_J = |P \cap T| / |P \cup T|$, where $| \ldots |$ denotes the set size and $\cap$ and $\cup$ correspond to intersection and union, respectively. Using alternative error measures, like the Hamming distance, produced similar results.

Deviations from a perfect ellipsoid, like more rectangular shapes or asymmetrical shapes, can reduce the match (*Figure 4—figure supplement 2*). Manual inspection of response fields revealed that most models with a Jaccard similarity $S_J > 0.5$ were well described by an ellipse, and we therefore chose this $S_J$ as a threshold for considering a response field well fitted by an ellipsoid. For the asymmetry index, we extracted the length of the major and minor axis of the best-fitted ellipse as the eigenvalues of the covariance matrix and took their ratio.

## Identification of preference types

Response fields were assigned to one of the four principal types in *Figure 1B* based on two inclusion criteria: first, the angle of the main axis had to fall within ± 10° of that of the prototypical angle (–45° for period, 0° for duration, 45° for duty cycle, 90° for pause). Second, the response field had to be sufficiently selective for the designated stimulus parameter and tolerant for the orthogonal stimulus parameter. For instance, a response field was determined to be pause tuned if the angle was ~90 ± 10°, and if it was selective for pause and tolerant for duration. We ensured the validity of these criteria through visual inspection of many response fields (see examples in *Figure 4E*). The orientation angle was calculated by fitting a line to the "ridge of the response field" (the set of pause and duration values that elicited maximal responses). We first identified well-responded stimuli as those with response values exceeding 50% of the maximal response value for that field. To make the fits more robust, we determined whether the set of well-responded stimuli extended more along the pause or the duration axis and selected pause and duration values for the fit as follows: if the response field was most extended along the pause axis, then we identified the preferred duration at each pause value for which the response field was above the 50% threshold. For a response field that extended more along the duration axis, we identified the preferred pause at each duration value exceeding the 50% threshold. We then fitted a line to the resulting set of duration and pause values and took the inverse tangent of that line's slope as the orientation angle.

## Acknowledgements

We thank Biswa Sengupta for discussions during an early phase of the project.

## Additional information

### Funding

| Funder | Grant reference number | Author |
|---|---|---|
| Biotechnology and Biological Sciences Research Council | BB/J01835X/1 | Berthold Hedwig Konstantinos Kostarakos |
| Royal Society | Newton International Fellowship | Konstantinos Kostarakos |
| Leibniz-Gemeinschaft | SAW 2012-MfN-3 | R Matthias Hennig |
| Deutsche Forschungsgemeinschaft | HE 2812/4-1 | R Matthias Hennig |
| Deutsche Forschungsgemeinschaft | HE 2812/5-1 | R Matthias Hennig |
| Deutsche Forschungsgemeinschaft | CL 596/1-1 | Jan Clemens |
| Deutsche Forschungsgemeinschaft | CL 596/2-1 | Jan Clemens |
| Deutsche Forschungsgemeinschaft | SCHO 1822/3-1 | Stefan Schöneich |

The funders had no role in study design, data collection and interpretation, or the decision to submit the work for publication.

### Author contributions

Jan Clemens, Conceptualization, Formal analysis, Writing - original draft; Stefan Schöneich, Konstantinos Kostarakos, Data curation, Writing – review and editing; R Matthias Hennig, Berthold Hedwig, Conceptualization, Supervision, Writing – review and editing

### Author ORCIDs

Jan Clemens (iD) http://orcid.org/0000-0003-4200-8097

Stefan Schöneich 🔟 http://orcid.org/0000-0003-4503-5111
Berthold Hedwig 🔟 http://orcid.org/0000-0002-1132-0056

**Decision letter and Author response**
Decision letter https://doi.org/10.7554/eLife.61475.sa1
Author response https://doi.org/10.7554/eLife.61475.sa2

## Additional files

### Supplementary files
• Transparent reporting form

### Data availability
The data used for fitting the model can be found at https://data.goettingen-research-online.de/dataverse/cricketnet. Code for running the model can be found at https://github.com/janclemenslab/cricketnet.

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
