## [Editor Report]

Clemens et al. present a computational model of the cricket song recognition network, which they show is capable of reasonably reproducing neural activity and song selectivity in *G. bimaculatus*. They then explore the parameter space of this network and find that varying parameters of model cells enable it to produce a range of selectivities for the period, pulse duration, duty cycle, or pause duration of input song. They then identify the network parameters that most affect song selectivity and investigate the relationship between several subsets of parameters and song preference. This is a fascinating exploration of the computational flexibility of a small neural circuit; it is well researched and written and was enjoyable to read.

---

## [Decision Letter]

**Decision letter after peer review:**

Thank you for submitting your article "A small, computationally flexible network produces the phenotypic diversity of song recognition in crickets" for consideration by *eLife*. Your article has been reviewed by 3 peer reviewers, and the evaluation has been overseen by Ronald Calabrese as the Senior and Reviewing Editor. The following individuals involved in review of your submission have agreed to reveal their identity: Ann Kennedy (Reviewer #1); Barbara Webb (Reviewer #2); Martin Paul Nawrot (Reviewer #3).

The reviewers have discussed the reviews with one another and the Reviewing Editor has drafted this decision to help you prepare a revised submission.

Summary:

Clemens et al., present a computational model of the cricket song recognition network, which they show is capable of reasonably reproducing neural activity and song selectivity in G. bimaculatus. They then explore the parameter space of this network and find that varying parameters of model cells enable it to produce a range of selectivities for the period, pulse duration, duty cycle, or pause duration of input song. They then identify the network parameters that most affect song selectivity and investigate the relationship between several subsets of parameters and song preference. This is a fascinating exploration of the computational flexibility of a small neural circuit; it is well researched and written and was enjoyable to read.

Essential Revisions:

There were several concerns that must be addressed before the paper can be accepted.; including justification/discussion of the generality (beyond crickets) of the insights gained.

1. Explanation of the core principle of function is not clear, particularly as it relates to whether the filter properties in the model have some plausible biophysical counterpart. The authors must present a knowledgeable discussion of the biophysical mechanisms that could generate the described filter functions particularly the seemingly long delays in the linear filter shapes.

2. The use of LN2 output instead of AN1 output for neurons downstream of AN1 seems like an unnecessary kluge and could have affected the results. "For simplicity and since AN1 and LN2 produce similar responses, we used the output of LN2_M in lieu of AN1_M responses for all neurons postsynaptic to AN1_M". How is it simpler? It sounds like a mistake in constructing the model. It could strongly affect the results, e.g. by making the excitatory and inhibitory inputs to LN4 more similar, as both are derived from LN2, instead of excitation from AN1 and inhibition from LN2.This should be fixed and the model reassessed.

Along the same lines the authors should discuss how they would translate the network output into behavioral output. Currently they seemingly compare behavioral response fields with their network output response field – there was considerable confusion about Figure 1B – but this has not been fully explained.

3. Several of the results in this paper, such as the anti-diagonal bands in Figure 3 and Figure 4b, seem to be a consequence of the fact that all simulated pulse trains had a fixed duration of 140ms. Is it reasonable to assume that pulse trains would have such a precisely fixed duration? Is LN4 also selective for pulse trains of a specific duration? If there isn't data on this, it would be an interesting thing to test with the model: if the input pulse train has one extra pulse or one fewer pulse, how are model LN4 responses affected? Does the model predict a preferred pulse train duration, or could it be the case that (for instance) longer pulse trains always lead to stronger LN4 activation?

Along the same lines, a cricket song consists of pulse-pause patterns, which are grouped within chirps that are separated by chirp pauses. The present paper ignores this and refers to the pulse-pause pattern filtering as 'song recognition'. This concern relates how the model would perform for an extra pulse.

4. One of the biggest discrepancies between data and model is the LN5 response at longer periods/pause durations (Figure 2b). Is it possible for the authors to comment on how this discrepancy might impact their other findings in the paper?

5. The distribution of model tolerances in Figure 4d seems surprising. It seems to suggest that the four categories of tolerances (period/duration/duty cycle/pause) do not correspond to distinct categories of models, but rather that the tolerance axis can take essentially any orientation. Is this true of other cricket species as well? If so, might it be that dividing tolerant axes into these four categories is misleading, in that it imposes discrete categories onto what is really a continuously varying signal?

6. We wonder how reasonable it is to describe all models as having a narrow "selective" axis and a broad "tolerant" axis as depicted in 4A. Among generated models where LN4m was responsive, was the LN4m response field always reasonably fit by a single ellipse, or did any models have more complex response fields?

7. The authors should discuss the previously described behavioral inter-individual variability within the species bimacultus in relation to the inter-species variability, which they cover with their parameter distributions.

These major points are further explained in the individual reviews*Reviewer #1:*

Clemens et al., present a computational model of the cricket song recognition network, which they show is capable of reasonably reproducing neural activity and song selectivity in G. bimaculatus. They then explore the parameter space of this network, and find that varying parameters of model cells enable it to produce a tremendous range of selectivities for the period, pulse duration, duty cycle, or pause duration of input song. They then identify the network parameters that most affect song selectivity, and investigate the relationship between several subsets of parameters and song preference. This is a fantastic exploration of the computational flexibility of a small neural circuit; it is very well researched and written, and was enjoyable to read. Although I had a few questions about the paper contents (see below), I believe that all of these can be addressed by the authors, upon which I would warmly recommend this paper for publication in *eLife*.

1) Several of the results in this paper, such as the anti-diagonal bands in Figure 3 and Figure 4b, seem to be a consequence of the fact that all simulated pulse trains had a fixed duration of 140ms. Is it reasonable to assume that pulse trains would have such a precisely fixed duration? Is LN4 also selective for pulse trains of a specific duration? If there isn't data on this, it would be an interesting thing to test with the model: if the input pulse train has one extra pulse or one fewer pulse, how are model LN4 responses affected? Does the model predict a preferred pulse train duration, or could it be the case that (for instance) longer pulse trains always lead to stronger LN4 activation?

2) Because of the number of layers and nonlinearities in this model, it is hard to picture what is happening under the hood to give rise to preferences for a particular period, duration, duty cycle, or pause in Figure 4. I found Figure 3 to be very helpful for the example of G. bimaculatus- would it be possible to generate similar plots for some of the models from Figure 4e, for comparison?

3) I was a bit surprised by the distribution of model tolerances in Figure 4d. This seems to suggest that the four categories of tolerances (period/duration/duty cycle/pause) do not correspond to distinct categories of models, but rather that the tolerance axis can take essentially any orientation. Is this true of cricket species as well? If so, might it be that dividing tolerant axes into these four categories is misleading, in that it imposes discrete categories onto what is really a continuously varying value?

4) On a related note, I found myself wondering how reasonable it is to describe all models as having a narrow "selective" axis and a broad "tolerant" axis as depicted in 4A. Among generated models where LN4m was responsive, was the LN4m response field always reasonably fit by a single ellipse, or did any models have more complex response fields? In addition to the orientation and preference of the selective axis, is there anything to be learned from looking at the width of the tolerant/selective axes, or the preference range of the tolerant axis?

5) One of the biggest discrepancies between data and model is the LN5 response at longer periods/pause durations (Figure 2b). Is it possible for the authors to comment on how this discrepancy might impact their other findings in the paper?*Reviewer #2:*

This paper contributes an interesting study of how parameter variation in a five-neuron network, closely based on identified neurons in the cricket, can establish different temporal tuning properties. The main application is to cricket song recognition; although the paper argues for more general insight into temporal recognition circuits, this is somewhat limited. Similarly, the argument for evolutionary relevance, as explaining how the diversity of cricket song might arise, would be more strongly supported either by showing potential 'pathways' of divergence (ideally through co-evolution models of production and recognition) or a clearer link from the model components to plausible biophysical mechanisms that could produce the relevant properties (e.g. specific filter shapes, especially where these are assumed to be comprised of multiple components within one neuron). As such, though the work is sound, it is of somewhat narrow interest.

Main contributions: the context of the work is that evolution of sensory preferences has been well explored at receptor level, but not yet for more complex stimulus properties, for which more sophisticated neural processing is needed to identify the preferred property in the signal. The main result of the paper is that different song preferences, as found across cricket species, can be obtained for different parameter settings within the same circuit, with biases in the frequency of preference types that match phenotypic diversity. The specific parameters/neural properties that produce qualitative differences in tuning (preference for period, duration or duty-cycle) are examined in more detail to provide some mechanistic insight into the circuit. This is approached in a very thorough manner, e.g., looking at each neuron's contribution and carrying out a full exploration followed by a sensitivity analysis to focus on the most important properties, and as such is of also of interest from a purely methodological point of view in neural modelling.

Substantive concerns:

1) As someone familiar both with cricket song recognition research and neural modelling, I had to work very hard to understand the circuit function from the presented description. The paper seems to assume the reader has very close familiarity with the papers by Kostarakos and Hedwig, 2012 and Schöneich et al., 2015 rather than giving a sufficiently clear account. E.g. in the introduction, the key concept is described as coincidence detection of delayed original input (AN1) and a "post-inhibitory rebound driven by the end of each sound pulse (LN5)". On the face of it, this seems to be a mechanism for pulse duration tuning, not period tuning, and it is unclear why "feature detector neuron LN4 integrates excitatory input from LN3 and inhibitory input from LN2, …sharpens its selectivity." From close inspection Figure 2, the mechanism for period selectivity appears to be 1) the timing of the rebound from LN5 from one syllable coincides with the onset of the next syllable 2) the response per syllable without this input decays for repeated syllables, and is further reduced in LN4 by inhibition with the same pattern as AN1. To some extent these phenomena are discussed later in the paper with reference to the effects of specific parameters, e.g. to increase the duration of the rebound from LN5; but it is difficult for the reader to follow without having the initial conceptual understanding of the original model.

2) The model seems relatively complex (multiple, somewhat arbitrarily chosen filters for each neuron, many parameters) and there is no discussion of whether it could be simplified while retaining the flexibility to be tuned to different song properties. Nor does the reader gain much insight into whether the parameters causing particular effects are plausible, or what might be the biophysical basis (this is discussed only for time delay variables) that could be subject to genetic modification.

3) The argument for insight into evolution from close examination of this network is not very convincing. Why would the existing network in one species be the "mother network" for other species? In the introduction, it is argued that "song recognition networks must be selective and modifiable to adapt to changing signal patterns" but the evolutionary drive seems more likely to be the opposite – the song should adapt to the recognition. Discussion of the co-evolution of production and recognition is very limited.*Reviewer #3:*

This model study nicely and exemplarily describes how, in a sensory system with highly limited neuronal resources, a small generic network with 5 neuron types can flexibly generate a variety of tuning properties, allowing for species-specific auditory mate-recognition. Building on their previous body of works, the authors here employ a phenomenological (i.e. non-mechanistic) rate-based feed-forward circuit model, fitted to accommodate known single-neuron input/output features. The model faithfully predicts the animals' (average) response behavior to parametrically controlled sensory stimuli. Targeted parameter modifications can tune the network for different auditory pulse patterns. The authors argue that such a flexible generic network motive could allow for evolutionary fast species separation.

1) The authors state "The neuronal circuit … has been revealed p.4 …". However, is there clear anatomic evidence for the explicit network wiring of 5 neurons and 6 connections? Does each of these neurons exist only once per hemisphere of any individual or are the authors referring to neuron types? Clear evidence should be referenced or missing evidence should be critically discussed.

2) The authors present a purely phenomenological model. How are these computations implemented biophysically? Which synaptic, cellular and network mechanisms are involved? Discussion of possible mechanisms and references to relevant works seems mandatory, in particular with respect to the (long) delays / rebound delay in the causal filters and the divisive normalization. Discussing adaptation for type AN1 and LN1 by means of either SFA (Nagel and Wilson, 2011, Nat Neurosci 14(2); Farkhooi et al., 2013, PLoS CB 9(10); Benda et al., 2003,2008) or short-term depressing synapses is straightforward. The phenomenological rate model has its own value. However, the argument that the authors did not aim at a biophysical implementation because ion channels and conductances are not known is not a good argument, this would have prevented 95% of published model studies.

3) The authors repeatedly refer to "song-recognition". However, the authors only investigate pulse sequences (Figure 1B) neglecting the impact of chirp tuning (e.g. Grobe et al., 2012, JEB 2015; Meckenhäuser et al., 2013, PloS one, 8; Clemens and Hennig, 2013). This needs to be discussed.

4) The authors argue their model "has the capacity to reproduce the behavioral preferences" (p.8) with reference to behavioral tuning in Figure 1B. They should make explicit that this refers to the congruence of the LN4 response field (Figure 3) and behavioral response field (Figure 1B) for G. bimaculatus; no attempt was made to explicitly model behavioral output. How could behavior be generated? What is the typical delay between song onset and behavioral response? What is known about the behavioral decision circuit? Please discuss possible mechanisms of behavioral decision making such as the previously suggested drift diffusion models (Hennig et al., 2014, Front. Physiol 5; Clemens et al., 2014, PNAS 11; Meckenhäuser et al., 2014, Front. Sys. Neurosci 8) and possible others?

5) Inter vs. intra-species variability. To my knowledge there is large inter-individual variability in female G.bimaculatus behavior (Grobe et al., 2012, Meckenhäuser et al., 2013) and the authors sate tenfold neuron parameter variation within species. However, they model only an average animal and do not mention behavioral variability at all. How could parameter variation for inter-individual variability in their network differ from inter-species variability? This should be discussed.

6) What is the critical test for the existence of a 'multi-purpose' circuit across species? Do the authors expect the same basic network topology across species and did they attempt to identify it (anatomically) in any species other than G. bimaculatus? Can they think of other methods of validation? This should be addressed in the Discussion.

7) Authors briefly mention various song preference patterns across species. It would be great to see specific examples of song patterns for a few species along with matching model tuning, e.g. in a supplemental figure, possibly together behavioral date / response diagrams.

8) The core result on multi-species covered in Figure 4 and text indicates all four "principal types" of response fields can be achieved by the model with reference to Figure 1B. Where does Figure 1B come from? Are we looking at sketches? Or behavioral response fields measured in crickets? Please make this clear and give references to the underlying data. Clemens and Hennig 2013 and Hennig et al., 2014 do not show these graphs. Ideally, the authors can reproduce exemplary experimental data from diverse cricket species for illustration.

---

## [Author Response]

Essential revisions:There were several concerns that must be addressed before the paper can be accepted.; including justification/discussion of the generality (beyond crickets) of the insights gained.

Beyond crickets many animal groups produce regular series of sound pulses directed at a receiver. If analyzed on the short time scale, there are four axes (pulse rate/period, pulse duration, pulse pause, duty cycle). It is a strength of the model that it identifies the main computations (forward excitation, suppression, rebound depolarization, coincidence detection, summation) needed to extract relevant information along these axes independent of the specific biophysical mechanism. In other words, the same computation may have more than one biophysical implementation.

Because of this generality (in terms of signal structure and modelling approach), our computational analyses produce hypotheses for how song preference may change from one type to another in different species and we now provide an example for that in the discussion. Since each model element has a straightforward biophysical implementation, we can at the same time propose specific changes in biophysical parameters that will lead to the desired changes in behavior.

We made the following edits to clarify these points:

Introduction:

“Repetitive patterns of short pulses that are organized in groups on a longer time scale are a common feature of acoustic signaling in insects, fish, and frogs (Baker et al., 2019; Carlson and Gallant, 2013; Gerhardt and Huber, 2002) and the processing and evaluation of these pulse patterns is therefore common to song recognition systems across species. […] Insights from insects where assumptions on physiological relevant parameters like synaptic strengths, delays and membrane properties of individual neurons can be made and systematically tested are therefore relevant for studies of pattern recognition systems and the evolution of acoustic communication systems in general.”

Discussion:

“The three computations that define model tuning – response suppression, post-inhibitory rebounds, and coincidence detection – occur across species and modalities. […] This could be implemented for example by reducing postsynaptic GABA receptors or by lowering the expression levels of voltage-gated potassium channels.”

1. Explanation of the core principle of function is not clear, particularly as it relates to whether the filter properties in the model have some plausible biophysical counterpart. The authors must present a knowledgeable discussion of the biophysical mechanisms that could generate the described filter functions particularly the seemingly long delays in the linear filter shapes.

We now explain the working principle in more detail in the introduction, based on the reviewers’ suggestions:

“In this species, the selectivity for a narrow range of pulse periods is created in a network of 5 neurons and 6 synaptic connections by combining a delay-line with a coincidence-detector (Figure 1D). […] Lastly, the feature detector neuron LN4 receives excitatory input from LN3 and inhibitory input from LN2, which sharpens its selectivity by further reducing responses to pulse patterns that do not produce coincident inputs to LN3. LN4’s selectivity for pulse patterns closely matches the phonotactic behavior of the females (Figure 1E).”

We also added a new section to the Discussion on the biophysical mechanisms generating each model element:

“How can these changes be implemented in a biological network? Although our phenomenological model is independent of a specific biophysical implementation, all model components have straightforward biophysical correlates. […] These examples demonstrate, that our phenomenological model has a straightforward, physiologically plausible implementations and therefore can propose experimentally testable hypothesis for transitions between types of behavioral preferences.”

2. The use of LN2 output instead of AN1 output for neurons downstream of AN1 seems like an unnecessary kluge and could have affected the results. "For simplicity and since AN1 and LN2 produce similar responses, we used the output of LN2_M in lieu of AN1_M responses for all neurons postsynaptic to AN1_M". How is it simpler? It sounds like a mistake in constructing the model. It could strongly affect the results, e.g. by making the excitatory and inhibitory inputs to LN4 more similar, as both are derived from LN2, instead of excitation from AN1 and inhibition from LN2.This should be fixed and the model reassessed.

We agree that our simplifying assumption that AN1 and LN2 provide similar inputs to the other neurons in the network could have affected our results, in particular since we removed one degree of freedom, which could restrict the distribution of responses produced by the network. We had done the simplification because we lacked data from AN1.

To address the concern, we now added AN1 recordings, refitted the model, and repeated all analyses (including the generation of millions of responses from models with randomized parameters). To replicate the adaptation across pulses in a train observable in the AN1 responses, we added an explicit divisive normalization step to AN1. We find that nearly all results obtained are in line with the original model. Importantly, the key results on the biases of the network and the role of delays and inhibition/adaptation can be replicated with the adjusted model. This highlights the robustness of our conclusions.

Fitting the new model changed some results in minor ways, without affecting our conclusions:

– The proportion of responsive and sensitive models dropped from ~⅓ to ~1/11. This is because AN1 forms the network’s input and new parameters added to model AN1 lead to more parameter combinations. As a result, very low or very high AN1 responses will lead to no responses or the same response to all patterns, respectively.

– The sensitivity of the model to changes in AN1 parameters is now reduced (Figure 4G). This is likely because of the addition of new parameters to fit the AN1 response dynamics – most of these parameters do not affect tuning.

– The ranking of parameters in the sensitivity analysis is different (synaptic delays are slightly less important but still rank highly). However, the general trends remain (Figure 4H) – parameters that were important in the old model are still important in the revised version. The correlation of sensitivity ranking when changing pairs of parameters vs individual parameters is slightly weaker but still substantial (Figure 4—figure supplement 1, before: correlation 0.95, now 0.88).

– The decrease in the preferred duty cycle in LN4_M_ with rebound delay was linear before. This curve has now a weakly decreasing slope, likely because of the adaptation added in AN1 (Figure 5D). In addition, the effect of duty cycle on the amplitude and latency of inputs to LN4_M_ is now slightly different but the trend of later and weaker excitation onto LN4_M_ with increasing duty cycle remains (Figure 5F).

– Trends in adaptation remain but addition of one more source of adaptation in AN1 makes the relationship between the number of adaptation/inhibition sources in the model and the preferred duty cycle more complex. However, the general trend, that more adaptation leads to lower preferred duty cycles, remains. We now show a simplified summary plot for this results in Figure 5I and moved the detailed plot to the Figure 5—figure supplement 3. In addition, small details in the response dynamics for trace for LN3_M_ and LN4_M_ in Figure 5G changed.

Overall, all these changes concern small details but do not affect any of the conclusions drawn from the model.

Along the same lines the authors should discuss how they would translate the network output into behavioral output. Currently they seemingly compare behavioral response fields with their network output response field – there was considerable confusion about Figure 1B – but this has not been fully explained.

We apologize for not making explicit the connection between model output and behavioral response fields. For the regular patterns used in this study and found in natural cricket songs, the translation from LN4 output to phonotaxis is rather simple – the firing rate over the chirp of LN4 is proportional to the phonotaxis value in *Gryllus bimaculatus*. Integrative processes over timescales that exceed the duration of the chirp shape insect decision making (Meckenhäuser 2014, DasGupta 2014) and these timescales can be characterized using artificial stimuli with conflicting information (Poulet and Hedwig 2005, Clemens 2014). We did not explicitly include them in the model, since the focus of our study was in pulse pattern recognition and these longer integration timescales do not change the responses to the regular pulse trains used in this study.

We now make explicit that we use the LN4_M_ firing rate over a chirp to predict phonotaxis in the Methods (P23, L814):

“Prediction of phonotactic behavior from network responses

For regular pulse trains, the firing rate of LN4 (number of spikes per chirp/chirp duration) predicts phonotactic behavior very well (Schöneich et al., 2015) (Figure 1E). We therefore use the firing rate of LN4_M_ to predict behavioral response fields from network output.”

And also discuss the existence of longer integration timescales in cricket phonotaxis in Results:

“To translate responses of the output neuron of the network – LN4_M_ – into phonotaxis, we used a simple model: The firing rate of LN4 is strongly correlated with the female phonotaxis in G. bimaculatus (Figure 1E, Schöneich et al., (2015)) and we therefore took LN4_M_‘s firing rate averaged over a chirp to predict phonotaxis from the model responses. Integrative processes over timescales exceeding the chirp are known to affect behavior in crickets and other insects (Poulet and Hedwig (2005), see also Meckenhäuser et al,. (2014); Clemens et al., (2014); DasGupta et al., (2014)). We omit them here since they do not crucially affect responses for the simple, repetitive stimuli typical for pulse trains produced by crickets*.”*

Regarding the response fields shown in Figure 1B: Initially, the response fields showed manually generated response fields. We have now replaced 3 of them with schematics derived from behavioral data of three different cricket species. The fourth – pause tuning – is not known in crickets and we kept the manually generated field. This is now clarified in the figure’s caption.

3. Several of the results in this paper, such as the anti-diagonal bands in Figure 3 and Figure 4b, seem to be a consequence of the fact that all simulated pulse trains had a fixed duration of 140ms. Is it reasonable to assume that pulse trains would have such a precisely fixed duration? Is LN4 also selective for pulse trains of a specific duration? If there isn't data on this, it would be an interesting thing to test with the model: if the input pulse train has one extra pulse or one fewer pulse, how are model LN4 responses affected? Does the model predict a preferred pulse train duration, or could it be the case that (for instance) longer pulse trains always lead to stronger LN4 activation?Along the same lines, a cricket song consists of pulse-pause patterns, which are grouped within chirps that are separated by chirp pauses. The present paper ignores this and refers to the pulse-pause pattern filtering as 'song recognition'. This concern relates how the model would perform for an extra pulse.

This question relates to the selectivity of *G. bimaculatus* for the chirp. *G bimaculatus* males produce song with a relatively fixed number of pulses per chirp – variation is small and amounts to the addition or removal of individual pulses (Schöneich and Hedwig 2012, Jacob and Hedwig 2016). *G. bimaculatus* females do show a chirp selectivity, but with a rather wide tuning and large inter-individual variability (Grobe et al., 2012). Accepted chirp durations range from 120 ms to 300 ms and thus the addition or removal of a single pulse does not make much of a difference to the phonotactic response. We confirm that our model is also robust to addition/removal of a single pulse in a new Figure 3—figure supplement 2.

Although it remains an interesting question whether the chirp selectivity known from behavioral studies could be part of the present model network, there is no electrophysiological data to test this question available. Such data would require extensive electrophysiological recordings with tests of combinations of different chirp durations and chirp pauses for several combinations of pulse-pause patterns as in Grobe et al., (2012). The *Gryllus* songs are in most cases songs with a chirped structure and as our manuscript focuses on fast time scale recognition of the pulse-pause pattern, we do indeed not assess all aspects of ‘song recognition’. The focus of our analyses on the short timescale of pulse duration and pause is now made explicit throughout the manuscript. For example in the abstract (P1, L22):

“Using electrophysiological recordings from the network that recognizes crucial properties of the pulse pattern on the short timescale in the cricket Gryllus bimaculatus, we built a computational model that reproduces the neuronal and behavioral tuning of that species. An analysis of the model’s parameter space reveals that the network can provide all recognition phenotypes for pulse duration and pause known in crickets and even other insects.”

In the introduction (P4, L107):

“We here asked whether the network that recognizes features of the pulse pattern on the short timescale in G. bimaculatus (Figure 1D) has the capacity to produce the diversity of recognition phenotypes for pulse duration and pause known from crickets and other insects (Figure 1B), and what circuit properties support and constrain this capacity. [..] By exploring the network properties over a wide range of physiological parameters, we show that the network of G. bimaculatus can be modified to produce all types of preference functions for pulse duration and pause known from crickets and other insect species.”

Results (examples):

– “We tested whether the delay-line and coincidence-detector network of the cricket G. bimaculatus (Figure 1D) can be modified to produce the known diversity of preference functions for pulse duration and pause in cricket calling songs (Figure 1B).” (P4, L120)

– Renamed section title from “The network can be tuned to produce all known song preferences of crickets” to “The network can be tuned to produce all known preferences for pulse duration and pause in crickets.” (P7, L212)

– We also used more precise wording when summarizing our results, for instance: “Our analysis of different model variants suggests that this song recognition network can produce all known preference types for pulse duration and pause over the range of stimulus parameters relevant for crickets.” (P9, L305)

We now also address the preference for chirp parameters as an open question and discuss possible aspects of chirp preference the network could contribute to (Discussion, P16, L515):

“In addition, we have not yet explored the network’s ability to reproduce the behavioral selectivity for parameters on the longer timescale of chirps (Figure 1A) (Grobe et al., 2012; Blankers et al., 2016; Hennig et al., 2016). It is likely that the network can explain some properties of the selectivity for chirp known from crickets. For instance, that a minimum of two pulses is required to produce coincidence in the network could at least partly explain the existence of a minimal chirp duration for G. bimaculatus (Grobe et al., 2012). Likewise, suppression in the network reduces responses to long chirps which could explain the reduced behavioral preference for long chirps. However, the current electrophysiological data do not sufficiently constrain responses at these long timescales and studies are needed to address this issue more comprehensively.”

4. One of the biggest discrepancies between data and model is the LN5 response at longer periods/pause durations (Figure 2b). Is it possible for the authors to comment on how this discrepancy might impact their other findings in the paper?

Although it remains an interesting question whether the chirp selectivity known from behavioral studies could be part of the present model network, there is no electrophysiological data to test this question available. Such data would require extensive electrophysiological recordings with tests of combinations of different chirp durations and chirp pauses for several combinations of pulse-pause patterns as in Grobe et al., (2012). The *Gryllus* songs are in most cases songs with a chirped structure and as our manuscript focuses on fast time scale recognition of the pulse-pause pattern, we do indeed not assess all aspects of ‘song recognition’. The focus of our analyses on the short timescale of pulse duration and pause is now made explicit throughout the manuscript. For example (relevant changes in bold) in the abstract (P1, L22):

“Using electrophysiological recordings from the network that recognizes crucial properties of the pulse pattern on the short timescale in the cricket Gryllus bimaculatus, we built a computational model that reproduces the neuronal and behavioral tuning of that species. An analysis of the model’s parameter space reveals that the network can provide all recognition phenotypes for pulse duration and pause known in crickets and even other insects.”

In the introduction (P4, L107):

“We here asked whether the network that recognizes features of the pulse pattern on the short timescale in G. bimaculatus (Figure 1D) has the capacity to produce the diversity of recognition phenotypes for pulse duration and pause known from crickets and other insects (Figure 1B), and what circuit properties support and constrain this capacity. [..] By exploring the network properties over a wide range of physiological parameters, we show that the network of G. bimaculatus can be modified to produce all types of preference functions for pulse duration and pause known from crickets and other insect species.”

Results (examples):

– “We tested whether the delay-line and coincidence-detector network of the cricket G. bimaculatus (Figure 1D) can be modified to produce the known diversity of preference functions for pulse duration and pause in cricket calling songs (Figure 1B).” (P4, L120)

– Renamed section title from “The network can be tuned to produce all known song preferences of crickets” to “The network can be tuned to produce all known preferences for pulse duration and pause in crickets.” (P7, L212)

– We also used more precise wording when summarizing our results, for instance: “Our analysis of different model variants suggests that this song recognition network can produce all known preference types for pulse duration and pause over the range of stimulus parameters relevant for crickets.” (P9, L305)

We now also address the preference for chirp parameters as an open question and discuss possible aspects of chirp preference the network could contribute to (Discussion, P16, L515):

“In addition, we have not yet explored the network’s ability to reproduce the behavioral selectivity for parameters on the longer timescale of chirps (Figure 1A) (Grobe et al., 2012; Blankers et al., 2016; Hennig et al., 2016). It is likely that the network can explain some properties of the selectivity for chirp known from crickets. For instance, that a minimum of two pulses is required to produce coincidence in the network could at least partly explain the existence of a minimal chirp duration for G. bimaculatus (Grobe et al., 2012). Likewise, suppression in the network reduces responses to long chirps which could explain the reduced behavioral preference for long chirps. However, the current electrophysiological data do not sufficiently constrain responses at these long timescales and studies are needed to address this issue more comprehensively.”

5. The distribution of model tolerances in Figure 4d seems surprising. It seems to suggest that the four categories of tolerances (period/duration/duty cycle/pause) do not correspond to distinct categories of models, but rather that the tolerance axis can take essentially any orientation. Is this true of other cricket species as well? If so, might it be that dividing tolerant axes into these four categories is misleading, in that it imposes discrete categories onto what is really a continuously varying signal?

It is true that the network can produce response fields of essentially any orientation. However, this is not true for cricket species, which cluster around the orientations corresponding to the four principal types. The division of the data into four tolerant axes is thus motivated by the diversity of the known behavioral preferences. Focusing on these types allows us to compare the biological diversity with that produced by the model. The fact that the model produces preferences at orientations not found in nature hints at additional selection pressures, like temperature robustness or environmental noise, that lead to the selection of the four types. However, we cannot rule out the existence of intermediate response types in so far uncharacterized species.

Based on this and the next point in Essential Revisions, we’ve modified our analysis and re-wrote this section. We now make our motivations more explicit and support it with references to behavioral studies that have characterized the response fields in various species of crickets and other insect species (Results):

“The preferred pulse parameters – duration, pause, and their combinations period, and duty cycle – only incompletely describe a network’s recognition phenotype. […] Last, duty-cycle tuning (Figure 1C, cyan) is given by diagonal alignment (θ=45^o^) and selectivity for duty cycle but tolerance for period (G. lineaticeps, Hennig et al., (2016)).”

We changed the histogram showing the distribution of orientations of the response fields from a log to a linear scale, to highlight the non-uniform distribution of response types (Figure 4D):

This shows that duration tuning is strongly over-represented, period and duty-cycle tuning are weakly enriched, and pause tuning is much rarer than expected. This unequal distribution of the orientations reflects the unequal distribution of preference types found in crickets. For instance, we know of no cricket species with pause tuning. This is now made clearer in a re-written section in Results (P9, L285):

“We find response fields with any orientation, again demonstrating that the network can produce more diverse response fields than has been reported in crickets. However, the orientations are unevenly distributed and are enriched for the principal types known from crickets: 36% of the response fields have an orientation of 0±10° which corresponds to duration tuning (expectation from uniform distribution: 20°/360° = 5.6%). Duty cycle tuning (45 ±10°) and period tuning (-45 ±10°) are also enriched, with 17% and 12%, respectively. Notably, pause tuning (90 ±10°) is not known in crickets and is the only principal type that is rarer than expected from a uniform distribution of orientations (2.0% vs. 2.8% expected). The rarity of pause tuning is consistent with the bias to prefer short pulse durations observed above (Figure 4C), since orientations around 90° requires response fields that extend parallel to the pulse duration axis. Note that these trends do not depend critically on the ranges of angles chosen for specifying different the response types.”

We also address possible implications of the fact that the model produces intermediate orientations which are not known from crickets (Results, P9, L300):

“Interestingly, the network tends to create a larger diversity of response fields than is known from crickets, for instance, fields that are symmetrical, multi-peaked, or have intermediate orientations. This suggests that biases in the network – like the rarity of pause tuning – constrain the distribution of preferences that evolution can select from, and that additional factors – like robustness to noise or temperature – then determine the ultimate distribution of phenotypes.”

6. We wonder how reasonable it is to describe all models as having a narrow "selective" axis and a broad "tolerant" axis as depicted in 4A. Among generated models where LN4m was responsive, was the LN4m response field always reasonably fit by a single ellipse, or did any models have more complex response fields?

To show that the response fields of most models are well described by single ellipsoids, we now characterize the response fields more systematically using three new analyses:

1. We determined whether a given response field was well described by an ellipsoid. We compute the overlap between each response field, binarized by thresholding at 50% of the field’s maximum and a best fitting ellipse. The overlap, given by the Jaccard similarity index is high for most models: the median Jaccard similarity is 0.83, 80% of all response fields have a Jaccard similarity >0.5. Manual inspection of response fields revealed that most models with a Jaccard similarity >0.5 are well described by an ellipse. We show a histogram of similarity values and examples of a high and a low overlap model in the new Figure 4—figure supplement 2A, B.

2. From the fitted ellipses, we computed an asymmetry index as the ratio between the length of the ellipses’ major and the minor axes. We find that most of the fields fitted well by a single ellipse (Jaccard similarity >0.5) are also asymmetrical (83% with an asymmetry index >1.25). The distribution of asymmetry indices is shown in a new Figure 4—figure supplement 2C.

3. We detected the number of distinct, well-separated peaks for each response field. “Dominant peaks were defined as local maxima in the response field exceeding 0.5 of the global maximum value and separated by a trough that is smaller than 0.75 of the value of the lower of the two peaks.” (Methods, P25, L929). This analysis revealed that the vast majority of fields (88%) has a single peak (Figure 4—figure supplement 2A). We now show examples from the 12% of the response fields with multiple peaks (new Figure 4—figure supplement 2D).

The results of these analyses are covered in a new Figure 4—figure supplement 2 and in new section in Results (P9, L275):

“We first examined to what extent the model produced the single-peaked, asymmetrical response fields typical for crickets. We find that most response fields (80%) produced by the selective model variants were well described by a single ellipse (Figure 4—figure supplement 2A, B, see Methods for details). Of these, 83% where asymmetrical (major axis >1.25x longer than minor axis), 17% were symmetrical (Figure 4—figure supplement 2C). 12% of all models produces multi-peaked response fields (Figure 4—figure supplement 2D), which are only known from katydids (Webb et al., 2007). The remaining 8% of the response fields were not well described by ellipses and/or did not have multiple distinct peaks. Thus, while the model produces more diverse responses – including complex, multi-peaked ones – most responses do match those typical for crickets.”

The analysis is also described in a new section “Characterization of response fields” in Methods:

“Characterization of response fields

The response fields known in crickets all have a single maximum and a roughly ellipsoid shape with a single maximum. […] For the asymmetry index, we extracted the length of the major and minor axis of the best-fitted ellipse as the eigenvalues of the covariance matrix and took their ratio.”

Taken together, the changes to how we present our results based on #5 and #6 in Essential Revisions provides a more complete and unbiased assessment of the diversity of responses produced by the model.

7. The authors should discuss the previously described behavioral inter-individual variability within the species bimacultus in relation to the inter-species variability, which they cover with their parameter distributions.

The behaviorally well-documented interindividual variability in phonotaxis behaviors (Grobe et al., 2012; Meckenhäuser et al., 2013) likely arises at multiple levels: At the level of song pattern recognition (interindividual differences in the network parameters), at the level of phonotaxis behavior (biases and noise in localizing the sound), at the motivational level (low or high motivation leads to more or less selective responses), and at the motor level (variability from motor noise). Identifying the contribution of variability at the level of song pattern recognition is challenging, since the full characterization of the behavioral phenotype in terms of the response fields cannot be obtained reliably at the individual level (the stimulus space is too large); this would require multiple testing of the same individuals with the same large set of test patterns. Therefore, our model of song pattern recognition does not explicitly consider the interindividual variability but is meant to represent the behavior of an average female.

Regarding the relation between intra vs inter species variability in network parameters: Our sensitivity analysis (Figure 4H) identified many parameters with little impact on the model tuning, which form one or multiple fitness-neutral subregions in parameter pace. In the biological population, these parameters likely vary between individuals, because they are subject to weak selection. By contrast, parameters with a high sensitivity score that affect crucial aspects of the recognition phenotype undergo stronger selection and are therefore likely less variable across individuals. These fitness-critical parameters are likely the ones driving changes in the recognition phenotype during a speciation event. Our current approach of assessing parameters individually (Figure 4G, H) or in pairs (Figure 4—figure supplement 1) is insufficient to capture the relationship of the fitness-neutral and the fitness-critical parameter subspaces Novel tools (for instance, Bittner et al., (2019) and Gonçalves et al., (2019)) will be employed to more exhaustively characterize the high-dimensional parameter to phenotype map with respect to inter and intra specific variability.

This point is now discussed in the discussion:

“We find that many parameters have only weak impacts on the network’s preference, leading to a relatively large fitness-neutral parameter set and a robust phenotype (Prinz et al., 2004). For instance, we observed that the rebound amplitude in LN5 tends to be variable across electrophysiological recordings in different individuals but that the network model’s output is relatively robust to these changes (Figure 2B, Figure 2—figure supplement 4). Other parameters crucially affect the recognition phenotype and support evolvability. These fitness-critical parameters likely drive changes in the recognition phenotype during speciation events to ensure species discrimination (Amézquita et al., 2011; Schmidt et al., 2011). Future studies will employ novel computational methods for characterizing the high-dimensional parameter space (Bittner et al., 2021; Gonçalves et al., 2020) to more comprehensively characterize the fitness-neutral parameter set leading to interindividual variability and to provide insight into the robustness and evolvability of pattern recognition networks.”

These major points are further explained in the individual reviewsReviewer #1 (Recommendations for the authors):Clemens et al., present a computational model of the cricket song recognition network, which they show is capable of reasonably reproducing neural activity and song selectivity in G. bimaculatus. They then explore the parameter space of this network, and find that varying parameters of model cells enable it to produce a tremendous range of selectivities for the period, pulse duration, duty cycle, or pause duration of input song. They then identify the network parameters that most affect song selectivity, and investigate the relationship between several subsets of parameters and song preference. This is a fantastic exploration of the computational flexibility of a small neural circuit; it is very well researched and written, and was enjoyable to read. Although I had a few questions about the paper contents (see below), I believe that all of these can be addressed by the authors, upon which I would warmly recommend this paper for publication in eLife.Major comments:1) Several of the results in this paper, such as the anti-diagonal bands in Figure 3 and Figure 4b, seem to be a consequence of the fact that all simulated pulse trains had a fixed duration of 140ms. Is it reasonable to assume that pulse trains would have such a precisely fixed duration? Is LN4 also selective for pulse trains of a specific duration? If there isn't data on this, it would be an interesting thing to test with the model: if the input pulse train has one extra pulse or one fewer pulse, how are model LN4 responses affected? Does the model predict a preferred pulse train duration, or could it be the case that (for instance) longer pulse trains always lead to stronger LN4 activation?

This is now addressed in detail in ‘Essential Revisions #3’.

Within a species, pulse trains (chirps) have a highly consistent duration – differences between chirps lie mainly in the addition or removal of a single pulse. Behavioral data show that for the chirp durations used here, female phonotaxis is robust to the addition/removal of a single chirp (Grobe, 2012). We performed a new analysis to show that this robustness to the addition or removal of a single pulse in a chirp is reproduced by the model (new Figure 3—figure supplement 2).

While the tuning for chirp parameters (chirp duration, chirp pause) is much broader than that for pulse parameters, crickets do indeed exhibit species-specific tuning for the chirp. However, the focus of this manuscript is on the tuning for pulse parameters, which we now make explicit throughout the text (see Essential Rev #3). Some aspects of the known behavioral tuning for chirp parameters are likely explained by the network but some aspects likely require additional mechanisms not captured by the pulse-pattern recognition network. This is now discussed (P16, L515, reproduced in Essential Rev #3).

2) Because of the number of layers and nonlinearities in this model, it is hard to picture what is happening under the hood to give rise to preferences for a particular period, duration, duty cycle, or pause in Figure 4. I found Figure 3 to be very helpful for the example of G. bimaculatus- would it be possible to generate similar plots for some of the models from Figure 4e, for comparison?

We now provide the response fields for all neurons in the network for an example from each response type in Figure 4E in a new Figure 4—figure supplement 3 in the supplement.

3) I was a bit surprised by the distribution of model tolerances in Figure 4d. This seems to suggest that the four categories of tolerances (period/duration/duty cycle/pause) do not correspond to distinct categories of models, but rather that the tolerance axis can take essentially any orientation. Is this true of cricket species as well? If so, might it be that dividing tolerant axes into these four categories is misleading, in that it imposes discrete categories onto what is really a continuously varying value?

This is now addressed in detail in ‘Essential Revisions #5’.

Known response fields from crickets fall into three categories – duration, period and duty cycle tuning. Pause tuning is not known. Intermediate types are not yet known. We have edited the text to clarify this. We have rewritten the relevant section in Results with new analyses and figures (Figure 4—figure supplement 2, 3) to provide a more nuanced assessment of the diversity of responses produced by the model.

4) On a related note, I found myself wondering how reasonable it is to describe all models as having a narrow "selective" axis and a broad "tolerant" axis as depicted in 4A. Among generated models where LN4m was responsive, was the LN4m response field always reasonably fit by a single ellipse, or did any models have more complex response fields? In addition to the orientation and preference of the selective axis, is there anything to be learned from looking at the width of the tolerant/selective axes, or the preference range of the tolerant axis?

This is now addressed in detail in ‘Essential Revisions #6’.

New analyses now more comprehensively describe the general shape of the response fields (new text in Results (P9, L275) and new Figure 4—figure supplement 2).

5) One of the biggest discrepancies between data and model is the LN5 response at longer periods/pause durations (Figure 2b). Is it possible for the authors to comment on how this discrepancy might impact their other findings in the paper?

This is now addressed in detail in ‘Essential Revisions #4’.

We now show that this discrepancy has only minor impact on the model responses (new text in Results, new Figure 2—figure supplement 4).

Reviewer #2 (Recommendations for the authors):This paper contributes an interesting study of how parameter variation in a five-neuron network, closely based on identified neurons in the cricket, can establish different temporal tuning properties. The main application is to cricket song recognition; although the paper argues for more general insight into temporal recognition circuits, this is somewhat limited. Similarly, the argument for evolutionary relevance, as explaining how the diversity of cricket song might arise, would be more strongly supported either by showing potential 'pathways' of divergence (ideally through co-evolution models of production and recognition) or a clearer link from the model components to plausible biophysical mechanisms that could produce the relevant properties (e.g. specific filter shapes, especially where these are assumed to be comprised of multiple components within one neuron). As such, though the work is sound, it is of somewhat narrow interest.Main contributions: the context of the work is that evolution of sensory preferences has been well explored at receptor level, but not yet for more complex stimulus properties, for which more sophisticated neural processing is needed to identify the preferred property in the signal. The main result of the paper is that different song preferences, as found across cricket species, can be obtained for different parameter settings within the same circuit, with biases in the frequency of preference types that match phenotypic diversity. The specific parameters/neural properties that produce qualitative differences in tuning (preference for period, duration or duty-cycle) are examined in more detail to provide some mechanistic insight into the circuit. This is approached in a very thorough manner, e.g., looking at each neuron's contribution and carrying out a full exploration followed by a sensitivity analysis to focus on the most important properties, and as such is of also of interest from a purely methodological point of view in neural modelling.Substantive concerns:1) As someone familiar both with cricket song recognition research and neural modelling, I had to work very hard to understand the circuit function from the presented description. The paper seems to assume the reader has very close familiarity with the papers by Kostarakos and Hedwig, 2012 and Schöneich et al., 2015 rather than giving a sufficiently clear account. E.g. in the introduction, the key concept is described as coincidence detection of delayed original input (AN1) and a "post-inhibitory rebound driven by the end of each sound pulse (LN5)". On the face of it, this seems to be a mechanism for pulse duration tuning, not period tuning, and it is unclear why "feature detector neuron LN4 integrates excitatory input from LN3 and inhibitory input from LN2, …sharpens its selectivity." From close inspection Figure 2, the mechanism for period selectivity appears to be 1) the timing of the rebound from LN5 from one syllable coincides with the onset of the next syllable 2) the response per syllable without this input decays for repeated syllables, and is further reduced in LN4 by inhibition with the same pattern as AN1. To some extent these phenomena are discussed later in the paper with reference to the effects of specific parameters, e.g. to increase the duration of the rebound from LN5; but it is difficult for the reader to follow without having the initial conceptual understanding of the original model.

This is now addressed in detail in ‘Essential Revisions #1’.

We explain the working principle in more detail in the introduction, based on the reviewers’ suggestions (P4, L94).

2) The model seems relatively complex (multiple, somewhat arbitrarily chosen filters for each neuron, many parameters) and there is no discussion of whether it could be simplified while retaining the flexibility to be tuned to different song properties. Nor does the reader gain much insight into whether the parameters causing particular effects are plausible, or what might be the biophysical basis (this is discussed only for time delay variables) that could be subject to genetic modification.

We agree that the number of parameters is large. This stems from our goal to reproduce the crucial aspects of the dynamics of all neurons faithfully. We believe that a high-dimensional and degenerate parameter space is something that our model shares with the biological network and it is key to the network’s computational flexibility. Our sensitivity analyses identified key parameters that shape crucial aspects of the model tuning (Figures 4, 5, 6). Finding a minimally parameterized circuit model that trades fidelity for simplicity while retaining computational capacity would be a fascinating topic for a future study!

These points are now addressed in Discussion:

“Computational flexibility also needs evolvability – the ability to generate specific and robust phenotypes during evolution (Blankers et al., 2015). The “mechanistic degeneracy” of computation in biological neural networks – the fact that specific network outputs can be obtained by tuning disparate parameters – is thought to be a prerequisite for evolvability (Wagner (2011), see also Hasson et al., (2020); Leonardo (2005); Schneider et al., (2021)). The degeneracy concept has been demonstrated extensively for motor networks (Gutierrez et al., 2013; Prinz et al., 2004), and here we also found evidence of degeneracy in the cricket song recognition network. For instance, the preferred pulse period (Figure 5A, B, Figure 5—figure supplement 1) or the pulse duty cycle (Figure 5G-I) can be set by multiple parameters. The hypothesis that high-dimensional and degenerate biological parameter spaces are more evolvable could be tested by assessing the computational flexibility of a minimal model of the network that produces the observed behavior with as few parameters as possible.”

To address the issue of the plausibility of our parameter values, we now discuss the possible biophysical bases for all parameters in the Discussion (P17, L543) (see ‘Essential Revisions #1’).

3) The argument for insight into evolution from close examination of this network is not very convincing. Why would the existing network in one species be the "mother network" for other species?

It is likely that large parts of the network are retained in closely related species and therefore that the network from G. bimaculatus resembles that found in closely related cricket species. This hypothesis is supported post-hoc by our results, which show that this network can produce all known behavioral preferences for pulse pause and duration in the species group. In addition, a recent phylogenetic analysis suggests that G. bimaculatus is close to the base of the phylogenetic tree from which many other species emerged (Gray et al., 2020, Page 17). Although this does not prove that the network of G. bimaculatus is a mother network for other species, phylogeny is in principle consistent with this idea.

This is now explained in the discussion:

“While the network model analyzed here is derived from recordings in one species (G. bimaculatus), the delay-line and coincidence-detector network is likely shared within the closely related cricket species. The phylogenetic position of G. bimaculatus close to the base of the phylogenetic tree from which many other species emerged is consistent with this idea (Gray et al., 2020). Our finding that this network can produce all known preferences for pulse and pause supports this idea and suggests that it for ms a common substrate – a “mother network” – for the diversity of song recognition phenotypes in crickets.”

In the introduction, it is argued that “song recognition networks must be selective and modifiable to adapt to changing signal patterns” but the evolutionary drive seems more likely to be the opposite – the song should adapt to the recognition. Discussion of the co-evolution of production and recognition is very limited.

This is true. The most likely scenario for the co-evolution of song preference and structure is the following: Female preferences drift around in signal space, maybe pushed by abiotic (environmental noise selects against preferences for very short pauses) and biotic factors (avoid overlap with heterospecifics (Amezquita, 2011)). Males then “adjust” their song to attract females. In this scenario, a female network that has the capacity to produce many different preference types supports the ability of communication system to diverge. Our finding that the network can produce all preference types confirms this hypothesis.

This is now clarified in Introduction:

“Since the evolution of song is largely driven by the female (Gray and Cade, 2000), the females’ song recognition must be selective and modifiable in order to drive the evolution of distinct, species-specific song patterns in males (Wagner, 2008).”

And in Ddiscussion:

“The computational flexibility of the recognition mechanism may explain the species richness as well as the speed of evolution in a particular taxon like crickets (Alexander, 1962; Blankers et al., 2015; Desutter Grandcolas and Robillard, 2003; Oh and Shaw, 2013; Otte, 1992): Female preferences drift around with little constraint in signal space, maybe pushed by abiotic (environmental noise selects against preferences for very short pauses) and biotic factors (avoid overlap with heterospecifics, Amézquita et al.,(2011)). The male song evolution follows female evolution since only males that sing attractive songs will reproduce. In this scenario, a female network that has the capacity to produce many different preference types supports the ability of the communication system to diverge. However, this co-evolution of song preference and song structure requires male song production networks to be as flexible as the female song recognition networks (Jacob and Hedwig, 2020; Schöneich, 2020). There is also accumulating evidence for genetic coupling between the networks that produce and recognize the song pattern, which may ensure that sender and receiver stay tuned during evolution of song pattern (Xu and Shaw, 2009; Schöneich, 2020; Xu and Shaw, 2021).”

Reviewer #3 (Recommendations for the authors):This model study nicely and exemplarily describes how, in a sensory system with highly limited neuronal resources, a small generic network with 5 neuron types can flexibly generate a variety of tuning properties, allowing for species-specific auditory mate-recognition. Building on their previous body of works, the authors here employ a phenomenological (i.e. non-mechanistic) rate-based feed-forward circuit model, fitted to accommodate known single-neuron input/output features. The model faithfully predicts the animals' (average) response behavior to parametrically controlled sensory stimuli. Targeted parameter modifications can tune the network for different auditory pulse patterns. The authors argue that such a flexible generic network motive could allow for evolutionary fast species separation.1) The authors state "The neuronal circuit … has been revealed p.4 …". However, is there clear anatomic evidence for the explicit network wiring of 5 neurons and 6 connections? Does each of these neurons exist only once per hemisphere of any individual or are the authors referring to neuron types? Clear evidence should be referenced or missing evidence should be critically discussed.

This is correct – the network is not based on a complete wiring diagram of the song recognition circuit in G. bimaculatus – it was reconstructed from anatomical data and single-cell recordings. We now refer to the different neurons in the network as cell types, of which multiple copies could exist per hemisphere.

This is clarified in Results:

“This network was previously inferred from the anatomical overlap together with the dynamics and the timing of responses of individually recorded neurons to a diverse set of pulse patterns (Kostarakos and Hedwig, 2012; Schöneich et al., 2015). Given that electrophysiology is challenging in this system, dual-electrode recordings to prove the existence of the inferred connections do not exist presently. We consider the neurons in the network cell types, that may also comprise multiple cells per hemisphere with highly consistent properties across individuals (Schöneich, 2020).”

2) The authors present a purely phenomenological model. How are these computations implemented biophysically? Which synaptic, cellular and network mechanisms are involved? Discussion of possible mechanisms and references to relevant works seems mandatory, in particular with respect to the (long) delays / rebound delay in the causal filters and the divisive normalization. Discussing adaptation for type AN1 and LN1 by means of either SFA (Nagel and Wilson, 2011, Nat Neurosci 14(2); Farkhooi et al., 2013, PLoS CB 9(10); Benda et al., 2003,2008) or short-term depressing synapses is straightforward. The phenomenological rate model has its own value. However, the argument that the authors did not aim at a biophysical implementation because ion channels and conductances are not known is not a good argument, this would have prevented 95% of published model studies.

This is now addressed in detail in ‘Essential Revisions #1’.

We added a detailed discussion of the biophysical implementation of all model elements to the discussion. We also now justify our phenomenological modelling approach more clearly (Results):

“We consider the neurons in the network cell types, that may also comprise multiple cells per hemisphere with highly consistent properties across individuals (Schöneich, 2020). We fitted a computational model based on intracellularly recorded responses of the network’s neurons to pulse trains. Our goal was to obtain a model that captures the computational capacity of the network without tying it to a specific biophysical implementation, we reproduced the responses of individual neurons using a phenomenological model based on four elementary computations (Figure 2A): 1. filtering, 2. nonlinear transfer functions (nonlinearities), 3. adaptation, and 4. linear transmission with a delay. This phenomenological model allows us to assess the network’s computational capacity independent of a specific biophysical implementation. However, all model components have straightforward biophysical correlates (see Discussion), which allows us to propose biophysical parameters that tune the network in specific implementations.”

3) The authors repeatedly refer to "song-recognition". However, the authors only investigate pulse sequences (Figure 1B) neglecting the impact of chirp tuning (e.g. Grobe et al., 2012, JEB 2015; Meckenhäuser et al., 2013, PloS one, 8; Clemens and Hennig, 2013). This needs to be discussed.

This is now addressed in detail in ‘Essential Revisions #3’.

This is correct, we focus on pulse sequences because selectivity for this aspect of the song is typically high while the tuning for chirp parameters tends to be broader. In addition, the network we model has been shown to explain pulse pattern recognition and no data exist that probe the network’s selectivity for long time scales. However, the network likely contributes to the preference for long timescales. This is now discussed (P16, L515) and we now make our focus on the recognition of pulse duration and pause explicit throughout the manuscript in abstract, intro, etc. (see ‘Essential Revisions #1’, reply P3-5).

4) The authors argue their model "has the capacity to reproduce the behavioral preferences" (p.8) with reference to behavioral tuning in Figure 1B. They should make explicit that this refers to the congruence of the LN4 response field (Figure 3) and behavioral response field (Figure 1B) for G. bimaculatus; no attempt was made to explicitly model behavioral output. How could behavior be generated? What is the typical delay between song onset and behavioral response? What is known about the behavioral decision circuit? Please discuss possible mechanisms of behavioral decision making such as the previously suggested drift diffusion models (Hennig et al., 2014, Front. Physiol 5; Clemens et al., 2014, PNAS 11; Meckenhäuser et al., 2014, Front. Sys. Neurosci 8) and possible others?

This is now addressed in detail in ‘Essential Revisions #2’.

For the regular patterns used in this study and found in natural cricket songs, the translation from LN4 output to phonotaxis is rather simple and we now make explicit that we use the firing rate of LN4 over a chirp as a predictor of phonotaxis (Results P8 L228, Methods P23, L817). We agree that integrative properties that exceed the duration of the pulse exist in insect decision making (Meckenhäuser, Dasgupta 2014) and that they can be revealed using artificial stimuli with conflicting information (Poulet 2005, Clemens 2014). However, these processes do not strongly affect responses to the repetitive stimuli used here. We discuss the existence of integrative processes in Results (P8, L231).

5) Inter vs. intra-species variability. To my knowledge there is large inter-individual variability in female G.bimaculatus behavior (Grobe et al., 2012, Meckenhäuser et al., 2013) and the authors sate tenfold neuron parameter variation within species. However, they model only an average animal and do not mention behavioral variability at all. How could parameter variation for inter-individual variability in their network differ from inter-species variability? This should be discussed.

See Essential Revisions #7.

6) What is the critical test for the existence of a 'multi-purpose' circuit across species? Do the authors expect the same basic network topology across species and did they attempt to identify it (anatomically) in any species other than G. bimaculatus? Can they think of other methods of validation? This should be addressed in the Discussion.

We have not done recordings or anatomical analyses in other species. We now propose behavioral and electrophysiological tests or our hypotheses in a new section in the discussion:

“How can the mother network hypothesis be tested? Behavioral tests can provide insight into whether other species use the coincidence detection algorithm found in G. bimaculatus (Hedwig and Sarmiento-Ponce, 2017). These experiments can for instance test the prediction that the duration of the last pulse in a chirp only weakly impacts network responses. Species that violate this prediction are unlikely to recognize song by the same coincidence mechanism. However, the “mother network” hypothesis does not imply that all crickets implement a coincidence detection algorithm, just that they reuse the same neurons with largely conserved response properties. In fact, our analyses have shown that through changes in key parameters, coincidence detection can be circumvented to produce a different preference type (Figure 6, Figure 4—figure supplement 1). That is why further electrophysiological experiments in G. bimaculatus are crucial to reveal the precise biophysical mechanisms that tune the network and ultimately link changes in gene expression, for instance, of specific ion channels, to changes in network tuning. Importantly, these experiments need to be extended to other species, by identifying and characterizing homologues of the neurons in the network. Recordings in other species are challenging but feasible, since homologous neurons are expected to be found in similar locations in the brain. Our model produces testable predictions based on the known behavioral tuning for how key properties of these neurons may look like in any given species (see below).”

7) Authors briefly mention various song preference patterns across species. It would be great to see specific examples of song patterns for a few species along with matching model tuning, e.g. in a supplemental figure, possibly together behavioral date / response diagrams.

We agree that an interesting next step would be to compare the model fitted to different species. We now show example response fields from three different species of crickets in Figure 1B and our analyses of the randomized models shows that the network can produce preferences for the pulse durations and pauses found in crickets (Figures 4B, C). However, as mentioned in Results, the preferred pulse-pause combination of the network is insufficient to describe the phenotype. Fitting the networks to the full response fields known from other species would be a large amount of work and is beyond the scope of the current, already quite long paper.

8) The core result on multi-species covered in Figure 4 and text indicates all four "principal types" of response fields can be achieved by the model with reference to Figure 1B. Where does Figure 1B come from? Are we looking at sketches? Or behavioral response fields measured in crickets? Please make this clear and give references to the underlying data. Clemens and Hennig 2013 and Hennig et al., 2014 do not show these graphs. Ideally, the authors can reproduce exemplary experimental data from diverse cricket species for illustration.

This is now addressed in ‘Essential Revisions #2’.

Figure 1B now shows response fields measured from crickets.